# Optogenetic dissection of basolateral amygdala contributions to intertemporal choice in young and aged rats

Caesar M Hernandez[1], Caitlin A Orsini[2], Chase C Labiste[1], Alexa-Rae Wheeler[1], Tyler W Ten Eyck[1], Matthew M Bruner[1], Todd J Sahagian[3], Scott W Harden[3], Charles J Frazier[3], Barry Setlow[2], Jennifer L Bizon[1]*

[1]Department of Neuroscience, University of Florida, Gainesville, United States; [2]Department of Psychiatry, University of Florida, Gainesville, United States; [3]Department of Pharmacodynamics, University of Florida, Gainesville, United States

**Abstract** Across species, aging is associated with an increased ability to choose delayed over immediate gratification. These experiments used young and aged rats to test the role of the basolateral amygdala (BLA) in intertemporal decision making. An optogenetic approach was used to inactivate the BLA in young and aged rats at discrete time points during choices between levers that yielded a small, immediate vs. a large, delayed food reward. BLA inactivation just prior to decisions attenuated impulsive choice in both young and aged rats. In contrast, inactivation during receipt of the small, immediate reward increased impulsive choice in young rats but had no effect in aged rats. BLA inactivation during the delay or intertrial interval had no effect at either age. These data demonstrate that the BLA plays multiple, temporally distinct roles during intertemporal choice, and show that the contribution of BLA to choice behavior changes across the lifespan.
DOI: https://doi.org/10.7554/eLife.46174.001

*For correspondence:
bizonj@ufl.edu

Competing interests: The authors declare that no competing interests exist.

## Introduction

Intertemporal choice refers to decisions between rewards that differ with respect to both their magnitude and how far in the future they will arrive. Biases in intertemporal choice, whether manifesting as extreme impulsivity or patience, strongly associate with psychiatric disease. For example, enhanced preference for smaller, immediate rewards (greater impulsive choice) is a hallmark of attention deficit hyperactivity disorder and substance use disorders (*Bickel et al., 2014*; *Hamilton et al., 2015*; *Patros et al., 2016*), whereas pronounced preference for delayed gratification is characteristic of the eating disorder anorexia nervosa (*Steinglass et al., 2012*; *Kaye et al., 2013*; *Decker et al., 2015*). Independent of psychopathology, intertemporal choice both associates with life outcomes and changes across the lifespan (*Denburg et al., 2007*; *Boyle et al., 2012*; *Beas et al., 2015*; *Hess et al., 2015*). Contrary to economic models predicting that older individuals should account for reduced time on the horizon in making intertemporal choices, healthy older adults actually exhibit a marked increase in preference for delayed outcomes (*Green et al., 1996*; *Green et al., 1999*; *Jimura et al., 2011*; *Löckenhoff et al., 2011*; *Mata et al., 2011*; *Samanez-Larkin et al., 2011*; *Eppinger et al., 2012*). Although this pattern of choice behavior is sometimes characterized as 'wisdom', increased preference for delayed over immediate rewards may also be maladaptive. For example, biases toward delayed gratification in older adults could contribute to inappropriately conservative financial strategies that forgo expenditures necessary to maintain quality of life.

The neural circuits underlying age-associated changes in intertemporal choice remain poorly understood. Relevant to elucidating this circuitry is the fact that intertemporal choice is a

**eLife digest** One marshmallow now or two in 15 minutes? That was the choice offered to young children in a classic psychology experiment known as the Stanford marshmallow test. Children who chose to wait went on to do better at school and to show healthier body weights in later life than those who ate the single marshmallow. A brain region called the basolateral amygdala (BLA) helps individuals choose between rewards that differ both in size and in when they will be available. Studies in people and in rodents show that the ability to wait for a larger reward – to delay gratification – increases with age. But whether changes in BLA activity contribute to this change was not known.

Choosing between a small reward now versus a larger one later involves several steps. Before a choice, individuals use their previous experience to compare the value of the immediate and the delayed rewards. How they feel at the time can bias this judgment. Someone who is hungry, for example, will assign greater value to receiving a single marshmallow now than someone who feels full. After making their choice, the individual then decides whether the reward they received was better or worse than they expected. This information helps them adjust their expectations for next time.

Hernandez et al. set out to examine how the BLA contributes to these different parts of the decision. Young and old rats were given a choice between a small food reward now or a larger reward after a delay. Hernandez et al. used optogenetic tools to temporarily inactivate the BLA either before or after the rats made their choice, and found that the role of the BLA varies across the lifespan. Inactivating the BLA before the choice made both young and old rats more likely to wait for the larger reward. By contrast, inactivating the BLA after a choice made young rats less likely to wait next time round, but had no effect in the older rats.

Changes in BLA activity with aging may thus make it easier to delay gratification in later life. But while the willingness of older adults to forego short-term rewards for long-term gain is often viewed as 'wisdom', such behavior can also be problematic. A pensioner who decides not to spend some of their savings on heating, for example, may be needlessly reducing their quality of life. Moreover, extreme impulsivity and extreme patience both feature in psychiatric disorders. The former may drive addiction, while the latter is a hallmark of anorexia. Identifying the mechanisms that underlie the ability to delay gratification may therefore help to promote effective decision-making in aging and psychiatric disorders.

DOI: https://doi.org/10.7554/eLife.46174.002

multicomponent process that involves a series of temporally distinct cognitive operations (*Rangel et al., 2008*; *Fobbs and Mizumori, 2017*). Specifically, most decisions begin with representations of past choices, as well as some idea of the outcomes associated with each choice option. These representations are weighted by one's motivation to obtain the choice outcomes at the time of the decision. A second phase of decision making occurs after a choice is made and involves evaluating the outcome to determine the degree to which it matches its predicted value. Feedback from this evaluative process can be used to adjust representations of the options to guide future choices. Both deliberation before a choice and outcome evaluation after a choice are supported by a network of brain structures that mediate reward processing, prospection, planning, prediction error, and value computations (*Peters and Büchel, 2011*; *Orsini et al., 2015a*; *Bailey et al., 2016*; *Fobbs and Mizumori, 2017*). The basolateral amygdala (BLA), which forms associations between cues or actions and their outcomes (*Johansen et al., 2012*; *Wassum and Izquierdo, 2015*), plays a central role in decision making and has been specifically implicated in both deliberative and evaluative processing (*Schoenbaum et al., 1998*; *Schoenbaum et al., 1999*; *Ghods-Sharifi et al., 2009*; *Peters and Büchel, 2011*; *Zuo et al., 2012*; *Grabenhorst et al., 2012*; *Zangemeister et al., 2016*; *Orsini et al., 2017*). The BLA also undergoes structural and functional alterations with advanced age, and BLA neural activity during intertemporal decision making is attenuated in aged rats (*Lolova and Davidoff, 1991*; *Rubinow and Juraska, 2009*; *Rubinow et al., 2009*; *Roesch et al., 2012*; *Burke et al., 2014*; *Prager et al., 2016*; *Samson et al., 2017*). It remains unclear, however, how age-associated changes in BLA recruitment actually influence intertemporal choice.

Optogenetic tools have been employed previously to define temporally-specific roles of BLA during deliberation and outcome evaluation in young rats performing a decision-making task involving risk of punishment (*Orsini et al., 2017*). Specifically, BLA inactivation at discrete timepoints in the decision process shifted choice behavior in opposite directions, highlighting multiple roles for BLA information processing in risky decision making. The present study used a similar optogenetic approach to define the roles of BLA neural activity in intertemporal choice (*Figure 1*) and to further determine if the roles of BLA change across the lifespan.

## Results

### Electrophysiological confirmation of light-induced inhibition of BLA neurons expressing eNpHR3.0.

Virally-transduced neurons were identified by mCherry expression and targeted for whole-cell patch clamp recordings using a combination of epifluorescence and differential interference contrast microscopy. Virally-transduced BLA neurons examined in slices from young and aged animals did not differ with respect to input resistance (Young: 122.6 ± 20.7 MΩ, n = 26 cells; Aged: 120.2 ± 11.6 MΩ, n = 28 cells; $t_{(52)}$=0.101, p=0.920), whole cell capacitance (Young: 139.7 ± 7.54 pF, n = 26 cells; Aged: 138.8 ± 6.91 pF, n = 28 cells; $t_{(52)}$=0.081, p=0.936), or current required to maintain the membrane potential at −70 mV (Young: −104.34 ± 15.9 pA; n = 26 cells; Aged: −116.0 ± 13.3 pA, n = 28 cells; $t_{(52)}$=0.566, p=0.574). Young and aged neurons filled with biocytin and visualized with 2-photon mediated epifluorescence microscopy were multipolar and had substantial dendritic branching, consistent with the morphology of BLA principal neurons (*Figure 2A,E*).

Light pulses (1 s duration) produced similar outward currents in young and aged virally-transduced BLA neurons, as observed in voltage clamp (young: 78.8 ± 10.6 pA, n = 26 cells, aged: 89.1 ± 8.4 pA, n = 28 cells, $t_{(52)}$=0.764, p=0.448, *Figure 2B,F*). Activation of eNpHR3.0 in this manner was consistently sufficient to silence both young and aged neurons when firing under a moderate (~50–200 pA) load (*Figure 2C,G*). Additional experiments were conducted to confirm that age differences did not emerge with longer light pulses (4 s) that matched or exceeded durations used in the in vivo experiments. A 4 s light-induced activation of eNpHR3.0 produced similar outward currents in young and aged cells (young: 55.2 ± 11.9 pA, n = 16 cells; aged: 55.9 ± 6.8 pA, n = 16 cells, $t_{(30)}$=0.050, p=0.961, *Figure 2D,H*). Finally, because long term activation of eNpHR3.0 plausibly could alter chloride gradients across the cell membrane, evidence for rebound excitation after 4 s light pulses was evaluated in both young and aged neurons. Overall, voltage clamp experiments measured in the first 500 msec after the light pulse revealed similar mean currents in young and aged neurons (young: 0.6 ± 1.6 pA, n = 16 cells, aged: 4.1 ± 2.1 pA, n = 16 cells, $t_{(30)}$=1.351, p=0.187). Similarly, only 1 of 61 neurons examined in current clamp (across both ages and light durations) that were silent before exposure to light fired any action potentials within 1 s of cessation of the light pulse. Overall, these data indicate that light-induced activation of eNpHR3.0 produces robust inhibition of virally-transduced BLA principal neurons, and that these effects do not significantly vary with age. The data further demonstrate that rebound excitation after eNpHR3.0 activation in BLA is minimal and unlikely to be functionally impactful.

### Fiber placement and AAV transduction

Expression of mCherry was used to confirm viral transduction in the BLA of rats used in behavioral studies that were injected with either AAV5-CamKIIα-eNpHR3.0-mCherry (AAV-eNpHR3.0, black circles in *Figure 3*) or AA5-CamKIIα-mCherry alone (AAV-control, white circles in *Figure 3*). Cannula placements were centered in the BLA, and the brain volumes virally transduced by AAV-eNpHR3.0 and AAV-control (calculated from the atlas of *Paxinos and Watson, 2005*) were comparable in young and aged rats.

### Effect of age on intertemporal choice performance

Previous work shows that aged rats display attenuated discounting of delayed rewards (*Simon et al., 2010*; *Hernandez et al., 2017*). Therefore, prior to inactivation sessions, delays were adjusted on an individual basis to ensure that all rats' choice performance was within the same parametric space (*Figure 4A*). This approach allowed a comparable range of effects from BLA

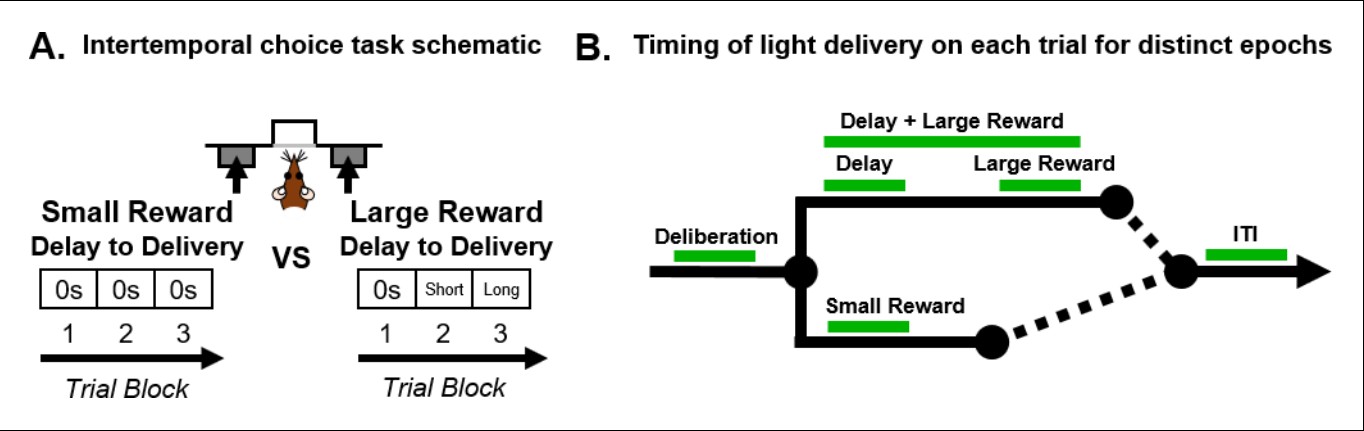

**Figure 1.** Schematics of intertemporal choice task and timing of light delivery. (**A**) Schematic of the intertemporal choice task illustrating the choices and trial blocks across which the duration of the delay to the large reward increased. On each trial, rats were presented with two response levers that differed with respect to the magnitude and timing of associated reward delivery. Presses on one lever delivered a small (one food pellet), immediate reward, whereas presses on the other lever delivered a large (three food pellets), delayed reward. Trials were presented in a blocked design, such that the delay to the large reward increased across successive blocks of trials in a session. (**B**) Schematic of a single trial in the intertemporal choice task showing the task epochs during which light was delivered (represented by the green line). Using a within-subjects design, light was delivered during *deliberation* (from when levers are presented until a choice is made); *small reward delivery; delay; large reward delivery; delay +large reward delivery;* and *intertrial interval* (*ITI*).

DOI: https://doi.org/10.7554/eLife.46174.003

inactivation to be observed in both young and aged rats, without concern for ceiling or floor effects. *Figure 4B* shows the actual delays used in the second and third blocks to achieve roughly 66% and 33% choice of the large reward, respectively, plotted as a function of age. A two-factor ANOVA (age × delay block) comparing the actual delays indicated the expected main effect of block ($F_{(2,26)}$=18.685, p<0.001, $\eta_p^2$=0.606, 1-β=0.930), as well as a main effect of age ($F_{(1,13)}$=6.402, p=0.025, $\eta_p^2$=0.330, 1-β=0.648) and an age × delay block interaction ($F_{(2,26)}$=6.913, p=0.004, $\eta_p^2$=0.347, 1-β=0.891). *Post hoc* analyses comparing the actual delays of young and aged rats in blocks 2 and 3 indicated that aged rats required longer delays than young to achieve comparable preference for large vs. small rewards (Block 2: $t_{(13)}$=-2.234, p=0.044, Cohen's *d* = 1.114, 1-β=0.480; Block 3: $t_{(13)}$=-2.660, p=0.020, Cohen's *d* = 1.328, 1-β=0.625). Consistent with this analysis, aged rats in comparison to young rats had a greater indifference point (the delay at which rats showed equivalent preference for large and small rewards; $t_{(13)}$ = −2.168, p=0.049, Cohen's *d* = 1.080, 1-β =0.457; *Figure 4C*).

## Effects on choice behavior of BLA inactivation during deliberation

Inactivation of the BLA during the deliberation epoch (n = 8 young and n = 7 aged) significantly increased choice of the large reward to the same extent in young and aged rats, particularly at long delays (*Figure 5A*). A three-factor ANOVA (laser condition × age × delay block) indicated a main effect of laser condition ($F_{(1,13)}$=103.507, p<0.001, $\eta_p^2$=0.888, 1-β=1.000) but no main effect of age ($F_{(1,13)}$= 0.089, p=0.770) nor an age ×laser condition interaction ($F_{(1,13)}$=1.838, p=0.198). A reliable main effect of delay block was observed ($F_{(2,26)}$=112.005, p<0.001, $\eta_p^2$=0.896, 1-β=1.000), as was as an interaction between laser condition and delay block ($F_{(2,26)}$=38.369, p<0.001, $\eta_p^2$=0.747, 1-β =1.000). Follow-up analyses, conducted to further explore the laser condition × delay block interaction, compared the effects of inactivation at each block. This analysis indicated that BLA inactivation significantly increased choice of the large reward in blocks 2 ($t_{(14)}$=-6.494, p<0.001, Cohen's *d* = 1.724, 1-β=0.995) and 3 ($t_{(14)}$=-9.434, p<0.001, Cohen's *d* = 2.228, 1-β=1.000), but not in block 1in which rats of both ages strongly preferred the large reward, even under control conditions ($t_{(14)}$=-0.323, p=0.751).

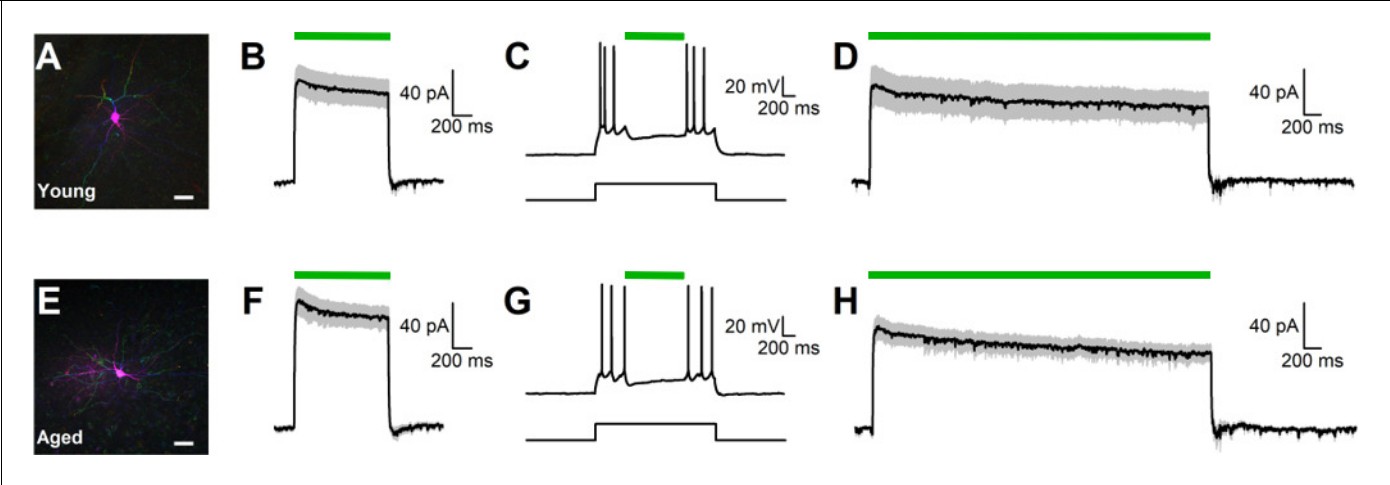

**Figure 2.** Functional inhibition of BLA pyramidal neurons via activation of halorhodposin. (**A**) A two-photon reconstruction of a biocytin-filled eNpHR3.0-expressing BLA neuron demonstrates multiple primary dendritic branches and spiny dendritic arborizations typical of BLA pyramidal neurons. Scale bar represents 20 μm. (**B**) Light-induced activation of eNpHR3.0 (1 s), indicated by the green line, produced a robust outward current in virally-transfected neurons in young rats when voltage clamped at −70 mV. Illustrated current is the average response observed in 26 BLA neurons from three young rats. (**C**) Representative trace shows that 1 s activation of eNpHR3.0 was reliably able to silence young neurons when firing under moderate load. The square pulse below the voltage trace for this cell indicates the time of current injection through the patch pipette that was sufficient to induce firing. The green line above the voltage trace indicates time of 1 s activation of eNpHR3.0. D: Light-induced activation of eNpHR3.0 (4 s) also produced a robust outward current in young BLA neurons, with no evidence of rebound excitation after termination of the light pulse. Illustrated current is the average response observed in 16 cells from two young rats. (**E-H**) Illustrate results of experiments identical to those presented in panels A-D, except in aged virally-transfected BLA neurons. Specifically, panel E shows a representative two-photon reconstruction of a virally-transducedaged BLA neuron (scale bar, 20 μm). Panel F illustrates average response to 1 s activation of eNpHR3.0 observed in 28 BLA neurons from three aged rats. Panel G illustrates a representative response to 1 s eNpHR3.0 activation during a suprathreshold current injection in a virally-transduced aged BLA neuron. Panel H illustrates the average response to 4 s activation of eNpHR3.0 observed in 16 BLA neurons from two aged rats. Overall, aging had no significant effect on intrinsic properties of BLA neurons, or on the effects of eNpHR3.0 activation in virally-transduced neurons. Raw data for electrophysiological analyses are provided in *Figure 2—source data 1*.

DOI: https://doi.org/10.7554/eLife.46174.004

The following source data is available for figure 2:

**Source data 1.** Hernandez et al. *Figure 2—source data 1*.
DOI: https://doi.org/10.7554/eLife.46174.005

## Effects on choice behavior of BLA inactivation during the small reward

In direct contrast to the effects of BLA inactivation during deliberation, BLA inactivation during the small reward epoch (n = 6 young and n = 6 aged) significantly decreased choice of the large reward only in young rats (*Figure 5C*). A three-factor ANOVA (laser condition × age × delay block) indicated main effects of laser condition ($F_{(1,10)}$=5.131, p=0.047, $\eta_p^2$=0.339, 1-β=0.534) and delay block ($F_{(2,20)}$=248.854, p<0.001, $\eta_p^2$=0.961, 1-β=1.000), but no interaction between laser condition and delay block ($F_{(2,20)}$= 1.317, p=0.290). Notably, although there was no main effect of age ($F_{(1,10)}$=0.941, p=0.355), the effects of BLA inactivation during small reward delivery did reliably interact with age (laser condition × age: $F_{(1,10)}$=7.127, p=0.024, $\eta_p^2$=0.416, 1-β=0.673). To better define the nature of this interaction, follow-up analyses using two-factor ANOVAs (laser condition × delay block) were performed on choice behavior separately in young and aged rats. BLA inactivation significantly decreased choice of the large reward in young rats (main effect of laser condition: $F_{(1,5)}$=18.226, p=0.008, $\eta_p^2$=0.785, 1-β=0.922, main effect of delay block: $F_{(2,10)}$=173.588, p<0.001, $\eta_p^2$=0.972, 1-β=1.000; laser condition × delay block: $F_{(2,10)}$=3.829, p=0.058) but not in aged rats (main effect of laser condition: $F_{(1,5)}$=0.061, p=0.814; main effect of delay block: $F_{(2,10)}$=93.015, p<0.001, $\eta_p^2$=0.949, 1-β=1.000; laser condition × delay block: $F_{(2,10)}$=0.185, p=0.834).

Because different delays to large reward delivery were required to achieve comparable levels of choice preference in young and aged rats, it is possible that the absence of BLA inactivation effects

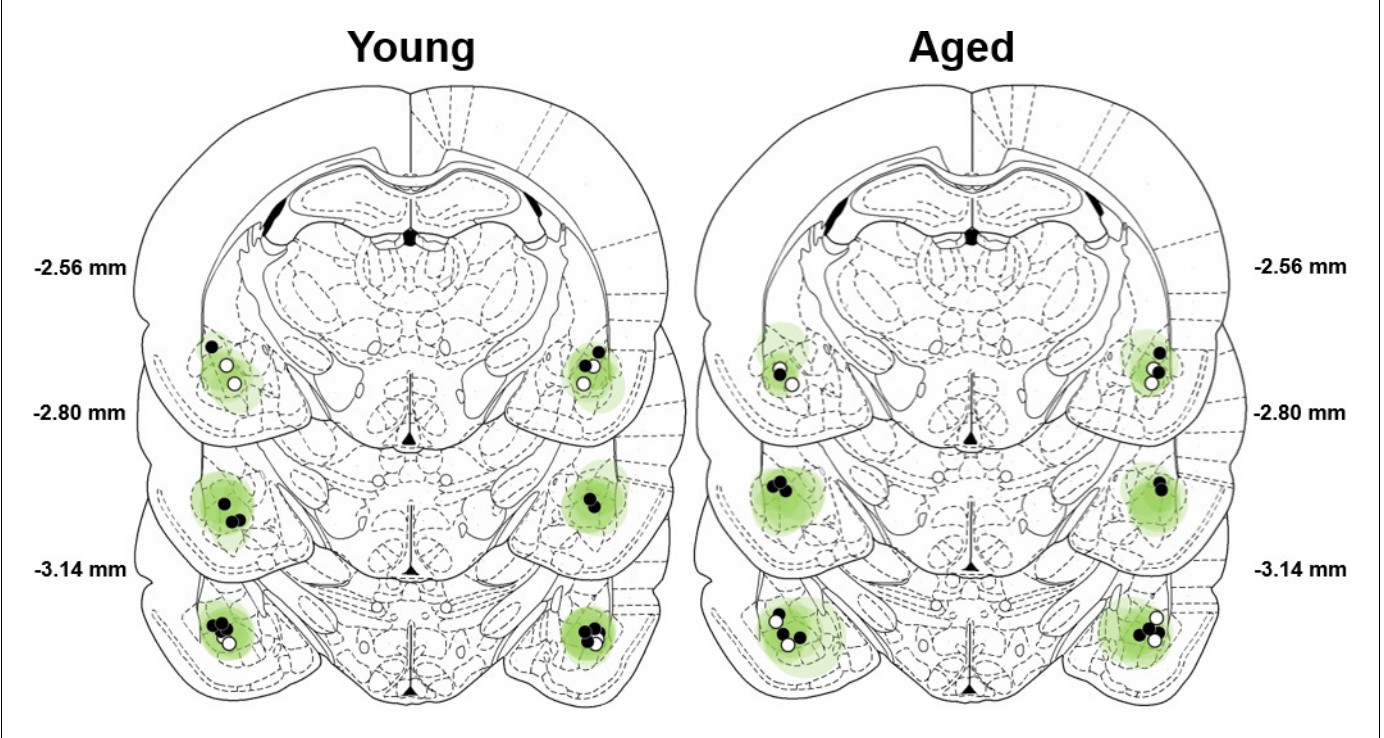

**Figure 3.** Verification of viral expression and fiber optic placements. The extent of viral-transduction in young (left) and aged (right) rats is depicted in green. Darker green indicates areas of greater expression of AAV5-CamKIIα-eNpHR3.0-mCherry or AA5-CamKIIα-mCherry(centered on the BLA), whereas lighter green indicates less expression (margins of the BLA). Filled black circles represent optic fiber placements in the experimental (eNpHR3.0) groups, and open circles represent optic fiber placements in the control groups. Viral expression and fiber placements are mapped to standardized coronal sections corresponding to −2.12 mm through −3.30 mm from bregma according to the atlas of *Paxinos and Watson (2005)*.

DOI: https://doi.org/10.7554/eLife.46174.006

in aged rats was due to the different delay durations employed rather than age differences per se. To address this possibility, an additional analysis was conducted in which aged rats were sub-grouped into those with delays comparable to young ('aged delay-matched') and those with delays that exceeded young ('aged delay-unmatched'). A multi-factor ANOVA was conducted with age subgroup (two levels: aged delay-matched and aged delay-unmatched) as the between-subjects factor and laser condition (two levels: laser on or off) and block (three levels: blocks 1, 2, and 3) as within-subjects factors. Critically, this analysis revealed no choice difference between aged delay-matched and aged delay-unmatched subgroups when the BLA was inactivated during the small reward delivery (main effect of sub-group: $F_{(1,4)}=0.180$, p=0.694; main effect of laser condition: $F_{(1,4)}=0.050$, p=0.834; main effect of block: $F_{(2,8)}=80.518$, p<0.001, $\eta_p^2=0.953$, 1-β=1.000; laser condition × sub-group: $F_{(1,4)}=0.082$, p=0.789; laser condition × block: $F_{(2,8)}=0.167$, p=0.849; sub-group × block: $F_{(2,8)}=0.328$, p=0.729; laser condition × sub group × block: $F_{(2,8)}=0.515$, p=0.616). These results indicate that it is unlikely that the different delay durations contributed to the age difference in the role of the BLA during small reward delivery.

## Altered choice strategy resulting from BLA inactivation during the deliberation and small reward epochs

The data above show that BLA inactivation during the deliberation and small reward epochs altered choice behavior in different directions (i.e., BLA inactivation during deliberation *increased* choice of the large reward in both young and aged rats, whereas BLA inactivation during small reward delivery *decreased* choice of the large, delayed reward in young rats but had no effect in aged rats). A trial-by-trial analysis was conducted on these data to determine the effects of BLA inactivation on two distinct behavioral strategies that could mediate these shifts in choice preference. Specifically,

during the deliberation epoch, this analysis determined the degree to which BLA inactivation influenced rats to 'shift' to the large reward option following a choice of the small reward on the previous trial, versus 'stay' with the large reward option following choice of the large reward on the previous trial. In the small reward epoch, the analysis assessed the degree to which BLA inactivation influenced rats to 'shift' to the small reward following choice of the large reward on the previous trial, versus 'stay' with the small reward following choice of the small reward on the previous trial.

As shown in *Figure 5B*, the percentage of trials during deliberation epoch inactivation on which a large reward choice was followed by a second large reward choice (large-stay) did not differ as a function of laser condition or age (main effect of laser condition: $F_{(1,13)}=2.563$, p=0.605; main effect of age: $F_{(1,13)}=0.282$, p=0.605; laser condition × age: $F_{(1,13)}=0.153$, p=0.702). In contrast, a similar analysis conducted on the percentage of trials on which a choice of the small reward was followed by choice of the large reward (small-shift) revealed a main effect of laser condition but no effect of age (main effect of laser condition: $F_{(1,13)}=40.051$, p<0.001, $\eta_p^2=0.755$, 1-β=1.000; main effect of age: $F_{(1,13)}=0.425$, p=0.526; laser condition × age interaction: $F_{(1,13)}=0.003$, p=0.954). Planned paired-samples t-tests showed that a significant increase in shifting after a choice of the small reward was evident in both young ($t_{(7)}=4.095$, p=0.005, Cohen's $d = 1.802$, 1-β=0.917) and aged ($t_{(6)}=5.342$, p=0.002, Cohen's $d = 2.442$, 1-β=0.987) rats. This finding indicates that the effects on choice behavior of BLA inactivation during deliberation result from rats shifting choices toward the large reward following a choice of the small reward.

Applying a parallel analysis to sessions in which inactivation took place during the small reward epoch yielded a different pattern of results (*Figure 5D*). BLA inactivation during the small reward epoch significantly increased the percentage of trials on which a small reward choice was followed by a second small reward choice (small-stay; main effect of laser condition: $F_{(1,10)}=6.026$, p=0.034, $\eta_p^2=0.376$, 1-β=0.601; main effect of age: $F_{(1,10)}=2.421$, p=0.151; laser condition × age interaction: $F_{(1,10)}=3.519$, p=0.090). Planned paired-samples t-tests showed that young rats were more likely to repeat the small reward choice on subsequent trials (small-stay; ($t_{(5)}=3.593$, p=0.016, Cohen's $d = 1.694$, 1-β=0.754) but this pattern was not observed in aged rats ($t_{(5)}=0.363$, p=0.732). In contrast, neither BLA inactivation nor age influenced the percentage of trials on which a choice of the large reward was followed by a choice of the small reward (large-shift; main effect of laser condition: $F_{(1,10)}=1.120$, p=0.315; main effect of age: $F_{(1,10)}=0.105$, p=0.753; laser condition × age: $F_{(1,10)}=0.086$, p=0.775).

## Effects of BLA inactivation during the deliberation and small reward epochs on other task performance measures

Other task measures were compared between BLA inactivation and baseline conditions in both deliberation and small reward epochs using a mixed-factor ANOVA, with age as the between-subjects factor and laser condition as the within-subjects factor. As shown in *Table 1*, the number of trials completed in a session did not differ as a function of laser condition or age in either the deliberation or small reward outcome epochs (Fs <3.431, ps >0.094). Similarly, as shown in *Table 2*, latency to press either the small or large reward lever did not differ as a function of laser condition or age in either epoch (Fs <4.149, ps >0.069). See *Tables 1* and *2* for full statistical results of these analyses.

## Effects of BLA inactivation during epochs associated with the large reward

Choosing the large reward lever resulted in a variable delay period that was followed by large (three food pellets) reward delivery. The effects of BLA inactivation during the delay and large reward delivery epochs were initially tested in separate sessions (n = 6 young and n = 6 aged). Subsequently, the effects of BLA inactivation across both the delay and large reward epochs were tested in a subset of these rats (n = 3 young and n = 3 aged).

### Effects of BLA inactivation during the delay epoch

The effects of BLA inactivation during the delay epoch were tested in delay blocks 2 and 3 using a three-factor ANOVA (laser condition × age × delay block). As expected, there was a main effect of delay block ($F_{(2,20)}=146.811$, p<0.001, $\eta_p^2=0.936$, 1-β=1.000) such that both young and aged rats

**Table 1.** Effects of BLA inactivation on number of trials completed per session.

| Epoch | Age | Laser condition | Mean | SEM | Statistical comparisons |
|---|---|---|---|---|---|
| Deliberation | Young | Off (Baseline) | 52.688 | 0.81 | Laser condition: $F_{(1,13)}=0.180$, p=0.678 |
| | | On (Inactivation) | 51.625 | 0.94 | Age: $F_{(1,13)}=0.162$, p=0.694 |
| | Aged | Off (Baseline) | 51.714 | 0.87 | Laser condition × Age: $F_{(1,13)}=3.264$, p=0.094 |
| | | On (Inactivation) | 53.429 | 1.003 | |
| Outcome (small reward) | Young | Off (Baseline) | 52.667 | 0.394 | Laser condition: $F_{(1,10)}=3.431$, p=0.094 |
| | | On (Inactivation) | 53.667 | 0.635 | Age: $F_{(1,10)}=0.180$, p=0.681 |
| | Aged | Off (Baseline) | 52.639 | 0.394 | Laser condition × Age: $F_{(1,10)}=0.328$, p=0.580 |
| | | On (Inactivation) | 53.167 | 0.635 | |

Raw data for this table are provided in **Table 1—source data 1**.
DOI: https://doi.org/10.7554/eLife.46174.011

The following source data is available for Table 1:
Source data 1. Hernandez et al. **Source data 1**.
DOI: https://doi.org/10.7554/eLife.46174.012

decreased their choice of the large reward as the delay prior to the large reward increased (**Figure 6A**). Compared to baseline, however, no reliable differences in choice behavior resulted from BLA inactivation during the delay epoch ($F_{(1,10)}=0.005$, p=0.947), nor was there an interaction between laser condition and delay block ($F_{(2,20)}=0.002$, p=0.998). Similarly, there were neither main effects nor interactions associated with age (main effect of age: $F_{(1,10)}<0.001$, p=0.996; age × delay

**Table 2.** Effects of BLA inactivation on lever response latencies.

| Epoch | Age | Lever | Laser condition | Mean (sec) | Std. error | Statistical analysis |
|---|---|---|---|---|---|---|
| Deliberation | Young | Large | Off (Baseline) | 1.385 | 0.174 | **Laser condition:** |
| | | | On (Inactivation) | 1.442 | 0.204 | *Large*: $F_{(1, 13)}=2.898$, p=0.112 |
| | | Small | Off (Baseline) | 1.083 | 0.187 | *Small*: $F_{(1, 8)}=2.050$, p=0.190 |
| | | | On (Inactivation) | 1.543 | 0.326 | **Age:** |
| | Aged | Large | Off (Baseline) | 0.956 | 0.186 | *Large::* $F_{(1, 13)}=1.988$, p=0.182 |
| | | | On (Inactivation) | 1.108 | 0.218 | *Small*: $F_{(1, 8)}=0.505$, p=0.497 |
| | | Small | Off (Baseline) | 1.043 | 0.187 | **Laser condition × Age:** |
| | | | On (Inactivation) | 1.120 | 0.326 | *Large:* $F_{(1, 13)}=0.588$, p=0.457 |
| | | | | | | *Small*: $F_{(1, 8)}=1.039$, p=0.338 |
| Outcome (small reward) | Young | Large | Off (Baseline) | 1.322 | 0.136 | **Laser condition:** |
| | | | On (Inactivation) | 1.217 | 0.141 | *Large*: $F_{(1, 10)}=1.429$, p=0.260 |
| | | Small | Off (Baseline) | 1.107 | 0.149 | *Small*: $F_{(1, 10)}=3.225$, p=0.103 |
| | | | On (Inactivation) | 0.938 | 0.054 | **Age:** |
| | Aged | Large | Off (Baseline) | 0.912 | 0.136 | *Large:* $F_{(1, 10)}=4.149$, p=0.069 |
| | | | On (Inactivation) | 0.870 | 0.141 | *Small:* $F_{(1, 10)}=1.157$, p=0.307 |
| | | Small | Off (Baseline) | 0.962 | 0.149 | **Laser condition × Age:** |
| | | | On (Inactivation) | 0.804 | 0.054 | *Large:* $F_{(1, 10)}=0.257$, p=0.623 |
| | | | | | | *Small:* $F_{(1, 10)}=0.004$, p=0.954 |

Raw data for this table are provided in **Table 2—source data 1**.
DOI: https://doi.org/10.7554/eLife.46174.013

The following source data is available for Table 2:
Source data 1. Hernandez et al.**Source data 1**.
DOI: https://doi.org/10.7554/eLife.46174.014

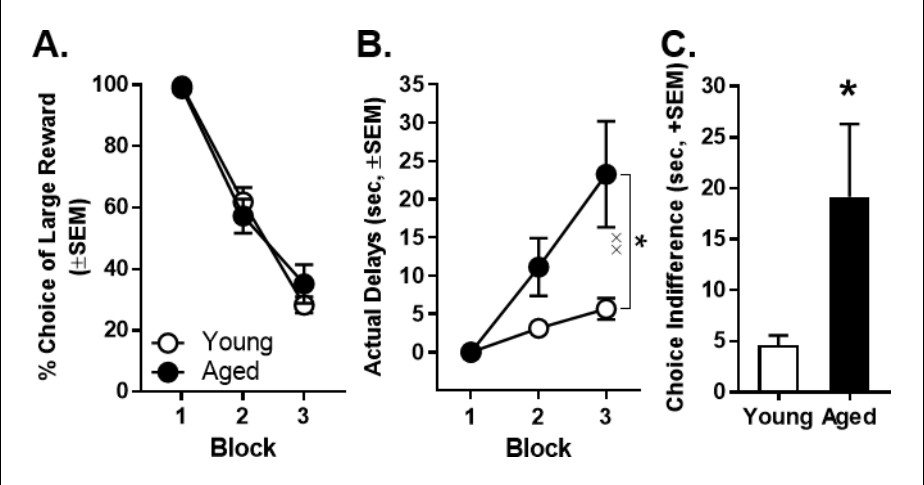

**Figure 4.** Effect of age on actual delays and indifference point. (**A**) Mean percent choice of the large reward in young and aged rats prior to initiation of BLA inactivation experiments. Note that delays to large reward delivery were adjusted individually for young (n = 8, open circles) and aged (n = 7, closed circles) rats in order to place all rats in the same parametric space. (**B**) Mean actual delays required to achieve the comparable young and aged rat choice performance shown in panel A. Aged rats required longer delays in Blocks 2 and 3 to achieve choice performance comparable to young rats. (**C**) The mean indifference point (the delay at which rats showed equivalent preference for the small and large rewards) was significantly greater in aged rats compared to young. In all panels, error bars represent the standard error of the mean (SEM). *p<0.05, main effect of age; ××p < 0.01, age × delay block interaction. Raw data for these graphs are provided in *Figure 4—source data 1*.

DOI: https://doi.org/10.7554/eLife.46174.007
The following source data is available for figure 4:

**Source data 1.** Hernandez et al. *Figure 4—source data 1*.

DOI: https://doi.org/10.7554/eLife.46174.008

block: $F_{(2,20)}=0.077$, p=0.926; laser condition × age: $F_{(1,10)}=0.081$, p=0.782; laser condition × age × delay block: $F_{(2,20)} = 0.096$, p=0.908).

## Effects of BLA inactivation during the large reward epoch

Unlike the effects of BLA inactivation during the small reward epoch, inactivation of BLA during the large reward epoch did not alter choice performance in either young or aged rats compared to baseline (*Figure 6B*). As expected, there was a main effect of delay block ($F_{(2,20)}= 120.846$, p<0.001, $\eta_p^2=0.924$, 1-β=1.000) such that both young and aged rats decreased their choice of the large reward as the delay to large reward delivery increased. Compared to baseline, however, no reliable differences in choice behavior resulted from BLA inactivation during the large reward epoch ($F_{(1,10)}=0.125$, p=0.731), nor was there an interaction between laser condition and delay block ($F_{(2,20)}=0.133$, p=0.876). Similarly, there were neither main effects ($F_{(1,10)}=0.249$, p=0.629) nor interactions associated with age (age × delay block: $F_{(2,20)}=0.437$, p=0.652; laser condition × age: $F_{(1,10)}=0.697$, p=0.423; laser condition × age × delay block: $F_{(2,20)}=0.664$, p=0.526).

## Effects of BLA inactivation during both delay and large reward epochs

One potential explanation for the null effects of BLA inactivation during the delay and large reward epochs is that, given the role of the BLA in integration of rewards and costs, inactivation may only be effective when conducted during both of these epochs. To evaluate this possibility, rats were tested while the BLA was inactivated during both the delay and large reward epochs; however, continuous inactivation across both epochs yielded no effects on choice performance. As shown in *Figure 6C*, a three-factor ANOVA (laser condition × age × delay block) revealed the expected main effect of delay block ($F_{(2,8)}=193.743$, p<0.001, $\eta_p^2=0.980$, 1-β=1.000) but no main effects or interactions involving laser condition or age (main effect of laser condition: $F_{(1,4)}=0.757$, p=0.433; main

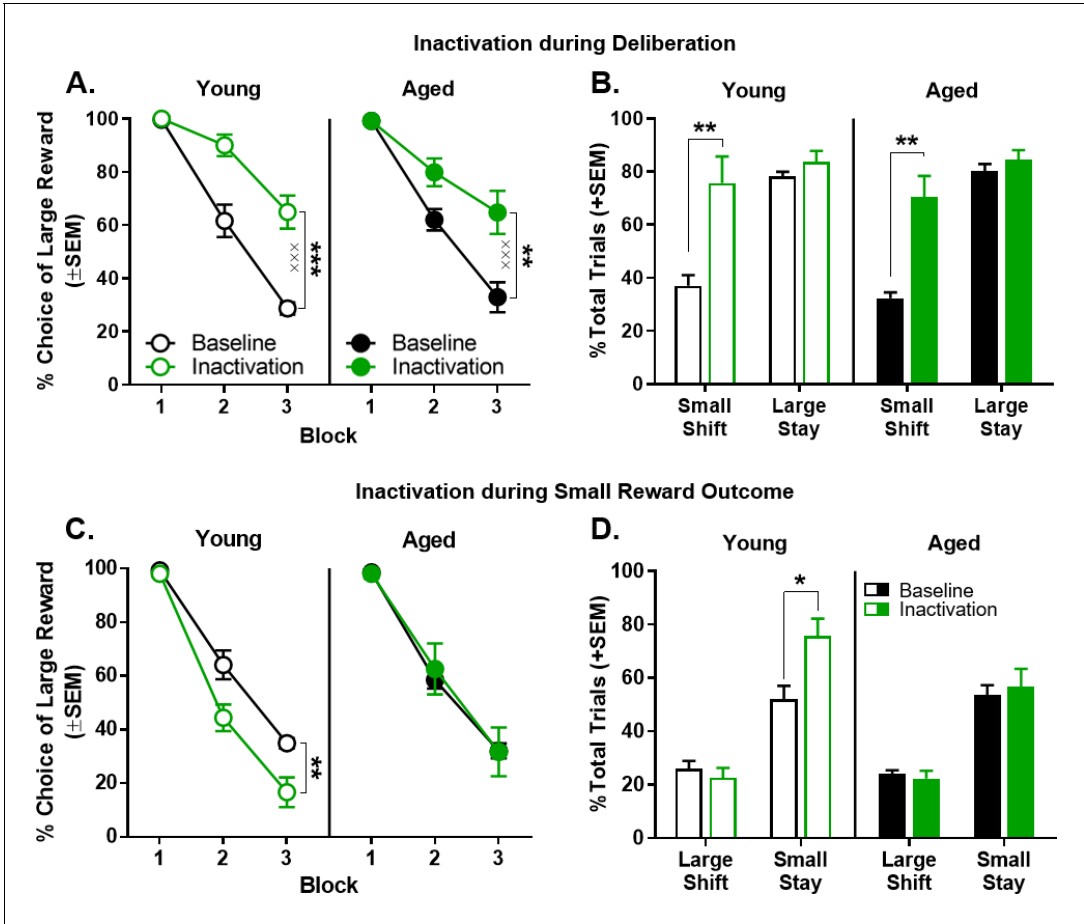

**Figure 5.** Effect of BLA inactivation during the deliberation and small reward epochs. (A) Inactivation of the BLA during the deliberation epoch (prior to a choice) resulted in a significant increase in preference for the large, delayed reward in both young (n = 8) and aged (n = 7) rats. (B) Effects of BLA inactivation during the deliberation epoch on trial-by-trial choice strategies. This analysis revealed that the increased choice of the large, delayed reward caused by BLA inactivation during deliberation (panel A) was due to an increase in the percentage of trials on which rats shifted to the large, delayed reward following a choice of the small, immediate reward. (C) Inactivation of the BLA during the small reward epoch resulted in a significant decrease in preference for the large, delayed reward in young (n = 6), but not aged (n = 6), rats. (D) Effects of BLA inactivation during the small reward epoch on trial-by-trial choice strategies. This analysis revealed that the decreased choice of the large, delayed reward in young rats caused by BLA inactivation during the small, reward epoch (panel C) was due to an increase in the percentage of trials on which rats 'stayed' on the small, immediate reward following a choice of this reward on the previous trial. In contrast, BLA inactivation during the same epoch in aged rats had no effect on trial-by-trial choice strategies. In all panels, error bars represent standard error of the mean (SEM). *p<0.05, **p<0.01, ***p<0.001, main effect of inactivation; ×××p < 0.001, inactivation × delay block interaction. Raw data for these graphs are provided in *Figure 5—source data 1*.

DOI: https://doi.org/10.7554/eLife.46174.009

The following source data is available for figure 5:

**Source data 1.** Hernandez et al. *Figure 5—source data 1*.

DOI: https://doi.org/10.7554/eLife.46174.010

effect of age: $F_{(1,4)}$=0.306, p=0.610; laser condition × delay block: $F_{(2,8)}$=0.979, p=0.417; laser condition × age: $F_{(2,8)}$=0.053, p=0.949; laser condition × age × delay block: $F_{(2,8)}$=0.159, p=0.856).

## Effects of BLA inactivation during the intertrial interval

To confirm the temporal specificity of the BLA inactivation effects, rats (n = 6 young, n = 6 aged) were tested while the BLA was inactivated during the intertrial interval (ITI). Although the expected main effect of delay block was observed ($F_{(2,20)}$=116.459, p<0.001, $\eta_p^2$=0.921, 1-β=1.000), BLA inactivation during the ITI did not alter choice performance compared to baseline in young or aged rats (main effect of laser condition: $F_{(1,10)}$=0.082, p=0.780; main effect of age: $F_{(1,10)}$=0.042, p=0.842;

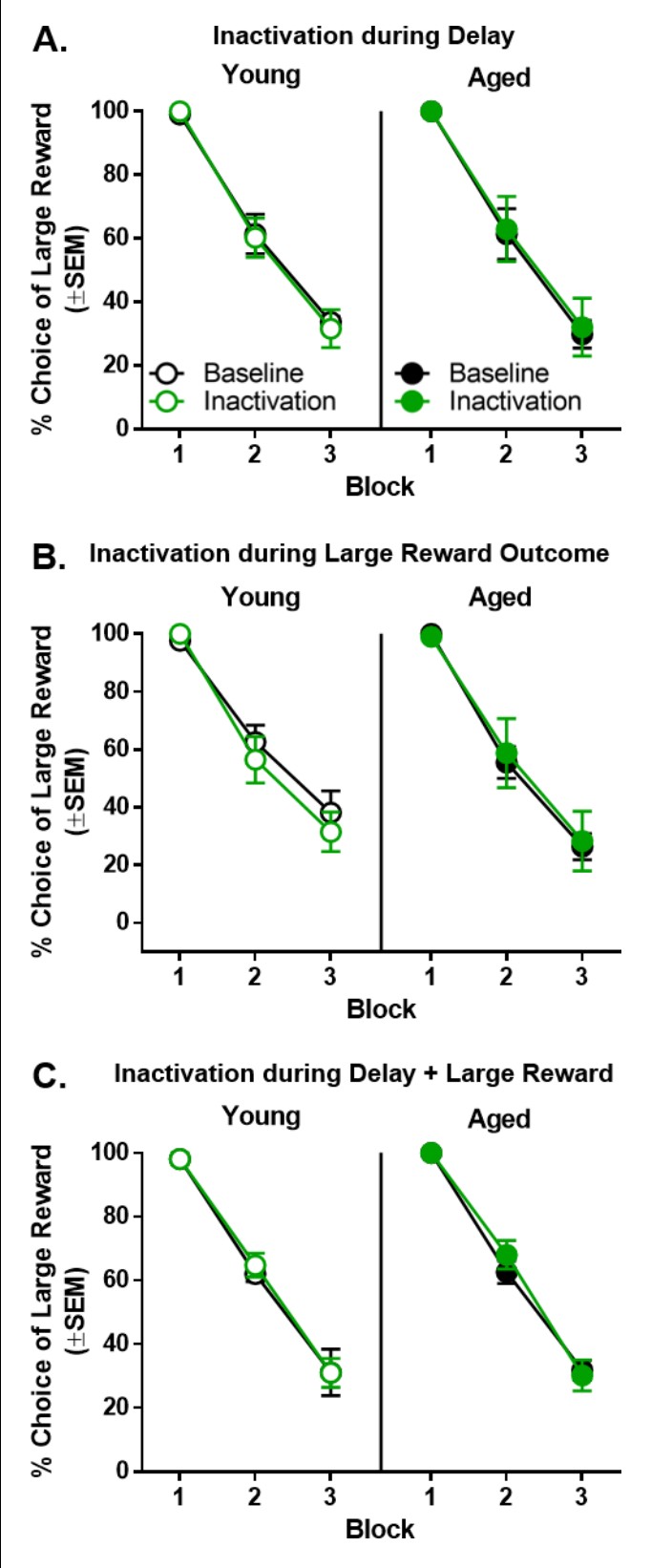

**Figure 6.** Effect of BLA inactivation during outcomes associated with choice of the large reward. (**A**) Inactivation of the BLA during the delay epoch resulted in no change in choice performance in either young (n = 6) or aged (n = 6) rats. (**B**) Inactivation of the BLA during the large reward epoch resulted in no change in choice performance

*Figure 6 continued on next page*

*Figure 6 continued*

in either young (n = 6) or aged (n = 6) rats. (C) Inactivation of the BLA during both the delay and large reward epochs resulted in no change in choice performance in either young (n = 3) or aged (n = 3) rats. Error bars represent standard error of the mean (SEM). Raw data for these graphs are provided in *Figure 6—source data 1*.
DOI: https://doi.org/10.7554/eLife.46174.015
The following source data is available for figure 6:

**Source data 1.** Hernandez et al. Figure 6 - source data 1
DOI: https://doi.org/10.7554/eLife.46174.016

laser condition $\times$ age: $F_{(1,10)}=0.298$, p=0.597; laser condition $\times$ delay block: $F_{(2,20)}=0.344$, p=0.713; age $\times$ delay block: $F_{(2,20)}=0.216$, p=0.808; laser condition $\times$ age $\times$ delay block: $F_{(2,20)}=0.198$, p=0.822; *Figure 7*).

## Effects of light delivery into BLA in rats with control virus (AAV5-CamKIIα-mCherry)

To control for non-specific effects of light delivery and viral transduction (e.g., changes in tissue temperature and off-target transduction effects), the effects of light delivery in rats virally transduced with a control virus that did not contain the eNpHR3.0 gene were tested during behavioral epochs in which BLA inactivation influenced choice behavior (i.e., deliberation: n = 4 young and n = 4 aged rats; and small reward: n = 4 young rats).

### Effects of light delivery during the deliberation epoch

Light delivery during the deliberation epoch in rats virally transduced with a control virus had no effects on choice performance (*Figure 8A*). A three factor ANOVA (laser condition $\times$ age $\times$ delay block) indicated the expected main effect of delay block ($F_{(2,12)}=100.272$, p<0.001, $\eta_p^2=0.944$, 1-β =1.000) but no main effects or interactions involving laser condition or age (main effect of laser condition: $F_{(1,6)}=0.128$, p=0.733; main effect of age: $F_{(1,6)}=0.055$, p=0.823; laser condition $\times$ age: $F_{(1,6)}=0.028$, p=0.874; laser condition $\times$ delay block: $F_{(2,12)}=0.121$, p=0.887; age $\times$ delay block: $F_{(2,12)}=0.105$, p=0.902; laser condition $\times$ age $\times$ delay block: $F_{(2,12)}=0.434$, p=0.658).

### Effects of light delivery during the small reward epoch

Light delivery during the small reward epoch in young rats transduced with control virus also failed to influence choice performance (*Figure 8B*). A two factor ANOVA (laser condition $\times$ delay block) indicated the expected main effect of delay block ($F_{(2,6)}=46.712$, p<0.001, $\eta_p^2=0.940$, 1-β=1.000) but no main effect of laser condition ($F_{(1,3)}=0.359$, p=0.592) or laser condition $\times$ delay block interaction ($F_{(2,6)}=0.173$, p=0.845).

## Discussion

Previous work employing permanent lesions or pharmacological approaches to manipulate BLA activity (*Winstanley et al., 2004*; *Churchwell et al., 2009*) has clearly demonstrated a critical role for this structure in intertemporal decision making. Specifically, BLA inactivation using such techniques results in more impulsive choices that favor immediate over delayed rewards. Lesion and pharmacological inactivation approaches, however, do not differentiate between temporally-discrete stages of decision making (e.g., deliberation, delay, outcome evaluation), which have been shown in other contexts to involve distinct neural circuitry (*Peters and Büchel, 2011*; *Fobbs and Mizumori, 2017*). The current study employed optogenetic tools in order to parse temporally-discrete contributions of BLA to intertemporal choice in both young and aged rats.

### Role of BLA in outcome evaluation in young rats

A large literature supports a role for BLA in assigning and updating the value of stimuli and events (*Hatfield et al., 1996*; *Málková et al., 1997*; *Baxter et al., 2000*; *Baxter and Murray, 2002*; *Shiflett and Balleine, 2010*; *Izquierdo et al., 2013*; *Parkes and Balleine, 2013*; *Wassum and Izquierdo, 2015*). With respect to decision making, the evaluative process mediated by BLA after an

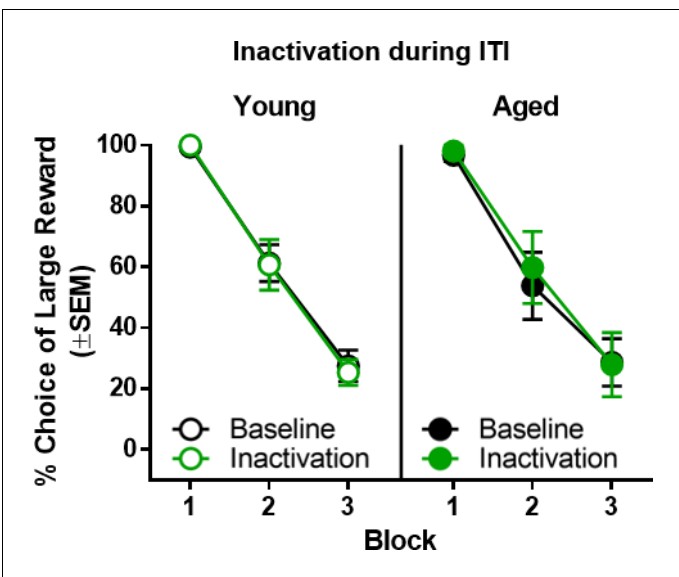

**Figure 7.** Effect of BLA inactivation during the intertrial interval. Inactivation of the BLA during the intertrial interval resulted in no change in choice performance in either young (n = 6) or aged (n = 6) rats. Error bars represent standard error of the mean (SEM). Raw data for these graphs are provided in *Figure 7—source data 1*.
DOI: https://doi.org/10.7554/eLife.46174.017

The following source data is available for figure 7:

**Source data 1.** Hernandez et al. *Figure 7—source data 1*.
DOI: https://doi.org/10.7554/eLife.46174.018

outcome has been received appears to involve acquisition and/or integration of information about the negative properties of that outcome. For example, previous work from our laboratory using a risky decision-making task demonstrated that BLA inactivation during receipt of a large, punished reward increased subsequent choice of this option over a small but safe reward (*Orsini et al., 2017*). Given this previous finding, it was somewhat surprising that in the current study, BLA inactivation during the large reward following the delay and/or the delay interval itself had no effect on choice behavior. These data demonstrate that the aversive properties of delays that bias choice behavior toward immediate options are not critically mediated by BLA. Importantly, however, BLA inactivation during the small reward epoch did reliably bias young rats toward choices of the small, immediate reward. This bias, which mimics that produced by BLA lesions (*Winstanley et al., 2004*; *Churchwell et al., 2009*), indicates that BLA is specifically important for evaluating and integrating the aversive properties that make the small reward less attractive than the large. Indeed, the trial-by-trial analysis showed that BLA inactivation rendered rats more likely to 'stay' with choices of the small reward after selection of that option on the previous trial, as though the negative feedback about that small reward 'not being good enough' had been attenuated. While these data are consistent with the idea that the BLA processes information about the aversive properties of outcomes in order to bias future behavior toward more favorable options (*Ghods-Sharifi et al., 2009*; *Orsini et al., 2017*), they also suggest that final integration of the values of both reward magnitude and delay occurs outside of the BLA. Given that working memory appears to contribute to the ability to delay gratification during intertemporal choice (*Shamosh et al., 2008*; *Bobova et al., 2009*; *Shimp et al., 2015*; *Hernandez et al., 2017*), it is likely that brain regions such as the hippocampus and prefrontal cortex mediate at least some components of information processing during the delay period. As such, one might predict that temporally-selective inactivation of these structures while waiting for a large reward (i.e., the delay interval) would influence future choice of that delayed option (*Churchwell et al., 2009*; *Mariano et al., 2009*; *Abela and Chudasama, 2013*; *Sonntag et al., 2014*; *Yates et al., 2014*). It should also be noted that BLA might be more critically involved in integrating information about delays and reward magnitude under other intertemporal choice conditions, such as in the presence of a cue that predicts large reward delivery during the

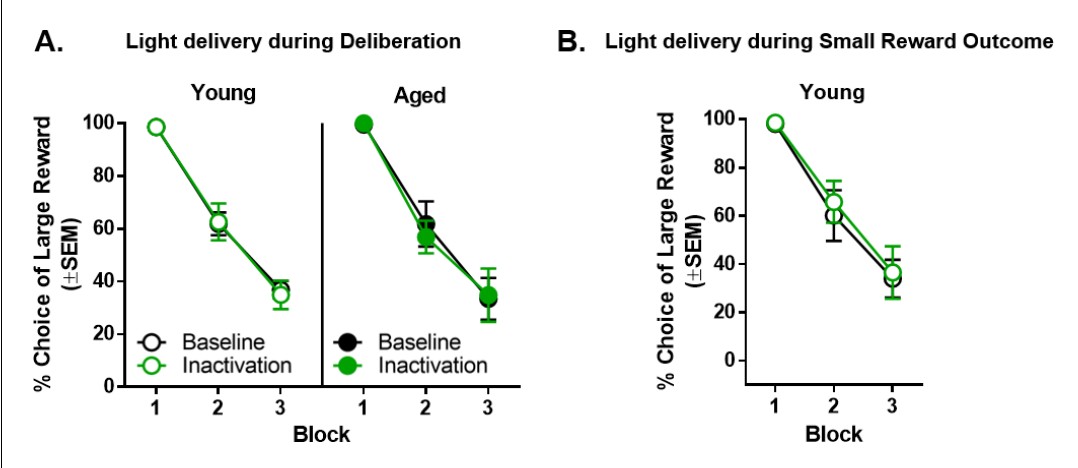

**Figure 8.** Effect of light delivery into BLA during the deliberation and small reward epochs in rats virally transduced with a control vector. (**A**) Light delivery into the BLA during the deliberation epoch resulted in no change in choice performance in either young (n = 4) or aged (n = 4) control vector rats. (**B**) Light delivery into the BLA during the small reward epoch resulted in no change in choice performance in young (n = 4) control vector rats. Error bars represent standard error of the mean (SEM). Raw data for these graphs are provided in *Figure 8—source data 1*.
DOI: https://doi.org/10.7554/eLife.46174.019

The following source data is available for figure 8:

**Source data 1.** Hernandez et al. *Figure 8—source data 1*.
DOI: https://doi.org/10.7554/eLife.46174.020

delay (*Zeeb et al., 2010*). Future work applying temporally-discrete inactivation approaches to other brain regions implicated in intertemporal decision making (e.g., prefrontal cortex and hippocampus) will help to more fully elucidate their unique and/or shared contributions with BLA to choice behavior.

## Role of BLA during deliberation in young rats

In contrast to results obtained during outcome evaluation, BLA inactivation during deliberation in young rats increased choice of the large, delayed reward. This shift toward less impulsive choice is opposite of that produced by inactivation methods such as lesions, which inhibit the BLA throughout the entire decision process (*Winstanley et al., 2004*; *Churchwell et al., 2009*). The prior study from our lab investigating risky decision making also showed that BLA inactivation during the deliberation epoch produced effects on choice behavior that were opposite those produced by neurotoxic lesions (*Orsini et al., 2015b*; *Orsini et al., 2017*). Specifically, during choices between small, safe and large, risky rewards, BLA lesions increase risky choice whereas selective optogenetic BLA inactivation during deliberation attenuates risky choice. Together, these data indicate a critical role for BLA during deliberation, which is normally overshadowed by its role in outcome evaluation. Importantly, although the risky decision-making and intertemporal choice tasks are analogous in design, performance on the two tasks is orthogonal and involves dissociable neural mechanisms (*Floresco et al., 2008*; *Churchwell et al., 2009*; *Simon et al., 2011*; *Mitchell et al., 2014*; *Orsini et al., 2015a*; *Orsini et al., 2018*; *Bailey et al., 2016*). Thus, the similar pattern of results observed with BLA manipulations suggests a common role for this structure in multiple forms of decision making. Specifically, BLA activity during deliberation under normal conditions may be important for ascribing incentive salience to the choice options, signaling their motivational value, or how much they are 'wanted', in the moment of the decision. Consequently, choice may be biased toward the more immediate reward in the intertemporal choice task, and toward the larger, albeit riskier, reward in the risky choice task. When this signal is removed (e.g., during BLA inactivation), rats are more likely to wait in the intertemporal choice task and less likely to risk punishment in the risky choice task to obtain the large reward. Indeed, the trial-by-trial analysis in the current study shows that BLA inactivation during deliberation renders rats more likely to shift their choices to the delayed reward following a choice of the immediate reward, as though the incentive properties

driving the immediate choice have been attenuated. This interpretation agrees with evidence from other behavioral contexts. For example, an intact BLA is necessary for the potentiating influence of reward-predictive cues on instrumental responding for reward (*Everitt et al., 2003*), as well as for maintaining effortful choices of preferred options (*Hart and Izquierdo, 2017*).

## Age differences in intertemporal choice

Across species, aging is accompanied by an increased ability to delay gratification (*Green et al., 1994*; *Green et al., 1999*; *Simon et al., 2010*; *Jimura et al., 2011*; *Löckenhoff et al., 2011*; *Samanez-Larkin et al., 2011*; *Eppinger et al., 2012*; *Hernandez et al., 2017*). Previous work from our labs showed that relative to young rats, aged rats display greater preference for large, delayed over small, immediate rewards in a 'fixed delays, block design' intertemporal choice task. This difference is not readily attributable to age-related deficits in cognitive flexibility, working memory, or food motivation, nor is it attributable to impairments in reward or temporal discrimination (*Simon et al., 2010*; *Hernandez et al., 2017*). The present study replicated these prior findings using a task variant in which the fixed delays/block design employed in our previous work was maintained, but the delays to large reward delivery were adjusted on an individual basis to obtain equivalent levels of choice preference in young and aged rats. Under these conditions, aged rats required longer delays to achieve levels of choice preference comparable to young, consistent with the idea that delays are less effective at discounting reward value in aged compared to young rats.

## Age differences in the role of BLA in intertemporal choice

Inactivation of BLA during the deliberation epoch decreased impulsive choice in both young and aged rats. In contrast, BLA inactivation during the small reward epoch increased impulsive choice in young rats but had no effect in aged rats. Importantly, this lack of effect in aged rats is unlikely to be attributable to age-related impairments in viral transduction or optogenetic efficacy. Histological reconstruction showed comparable BLA viral transduction in young and aged rats (*Figure 3*). Moreover, in vitro electrophysiological experiments showed that halorhodopsin-transduced BLA neurons in both age groups were silenced to a similar degree in response to light pulses (*Figure 2*). Most importantly, inactivation during the deliberation epoch in the same aged rats used to test the effects of inactivation during the small reward epoch produced effects on behavior that were as robust as those in young rats (*Figure 5A*). These latter data provide direct, in vivo verification that the absence of behavioral effects following BLA inactivation during the small reward epoch in aged rats cannot be ascribed to attenuated halorhodopsin efficacy. It is possible that aged rats' bias toward the large, delayed reward could have resulted in insufficient parametric space in which to observe optogenetically-induced shifts in choice behavior. The intertemporal choice task was explicitly designed to address this possibility by adjusting the delays to equate baseline choice preference in young and aged rats (*Figure 4A*). Additional analyses were conducted to ensure that the age difference in the delay to large reward delivery that resulted from this design did not itself influence the role of BLA in intertemporal choice. Specifically, aged rats were divided into subgroups based on whether their delays to large reward delivery matched or exceeded the range of young. The effects of BLA inactivation were identical between these aged subgroups, and in particular, BLA inactivation during the small reward epoch had no effect in either subgroup. These data indicate that the different delay durations experienced by young and aged rats do not account for the age differences in the role of BLA in intertemporal choice.

The distinct effects of BLA inactivation in young and aged rats could suggest that aged subjects fundamentally make decisions differently than young, relying less heavily on evaluation of choice outcomes (*Löckenhoff et al., 2011*; *Mather et al., 2012*; *Samanez-Larkin and Knutson, 2015*; *Pachur et al., 2017*). For example, whereas young adults readily incorporate new information to guide decisions, older adults tend to rely more heavily on previously-learned, 'crystalized' knowledge for decision making (*Horn, 1982*; *Mather, 2006*; *Mata et al., 2011*). If the cognitive structure of the decision process differs, the absence of BLA inactivation effects during outcome evaluation in aged rats may not necessarily reflect BLA dysfunction. For example, the increased reliance of older adults on crystalized knowledge for decision making has been attributed in part to vulnerability in neural structures such as hippocampus, which compromises encoding of new information and prospective memory (*Del Missier, 2015*; *Shadlen and Shohamy, 2016*; *Hu et al., 2017*; *Levin et al.,*

2017). Moreover, there is substantial evidence in aged humans and rodents for recruitment of brain circuits that are distinct from those engaged by young subjects during complex cognitive operations, even when performance is equated (*Antonenko and Flöel, 2014*; *Lighthall et al., 2014*; *Tomás Pereira et al., 2015*; *Wang et al., 2015*). The cause-effect relationships in the influences of aging on neural versus cognitive 'restructuring' are difficult to disentangle. It is certainly possible, however, that the fact that aged rats don't use the BLA for outcome evaluation during intertemporal decision making reflects neural dysfunction and/or compensation associated with other brain regions.

While the current findings do not necessarily reflect age-associated neurobiological impairments within the BLA, prior evidence does suggest that this region is susceptible to age-associated changes (*Rubinow and Juraska, 2009*; *Rubinow et al., 2009*; *Roesch et al., 2012*; *Prager et al., 2016*). Indeed, although BLA neuron number remains relatively stable with age, baseline firing rate of BLA neurons in vivo is reduced in aged rats (*Almaguer et al., 2002*; *Roesch et al., 2012*), and a recent electrophysiological recording study reported enhanced β-power in BLA of aged rats during reward evaluation in a probabilistic decision-making task (*Samson et al., 2017*). Projections from the BLA to the nucleus accumbens (NAc) may be particularly relevant for reward outcome evaluation. Pharmacological disconnection of the BLA and NAc impairs discrimination between a devalued vs. a non-devalued food reward (*Shiflett and Balleine, 2010*), and optogenetic inactivation of BLA terminals in NAc during outcome evaluation increases preference for the 'risky' option in a probabilistic decision-making task (*Bercovici et al., 2018*). These results support the idea that BLA projections to NAc are responsible for mediating negative feedback regarding choice outcomes. Notably, *Eppinger et al. (2013)* showed blunted activity in ventral striatum during reward prediction errors in older adults performing a learning task. Together with the current study, these findings suggest that the BLA-NAc circuit is disengaged during decision making in older adults.

Unlike BLA inactivation during outcome evaluation, inactivation during deliberation in both young and aged rats mimicked the attenuated impulsive choice observed in aging (*Figure 4C*; *Simon et al., 2010*; *Hernandez et al., 2017*). This effect is only observed using temporally-discrete optogenetic inhibition during deliberation, and not during outcome evaluation or with experimental methods such as lesions or pharmacological inactivation that inhibit the BLA across all stages of the decision process (*Winstanley et al., 2004*; *Churchwell et al., 2009*). Such data suggest that in young rats, activity in BLA circuits involved in outcome evaluation may be the primary driver of choice behavior. The failure to engage such outcome evaluation circuits in aging, however, may 'unmask' the contributions of BLA during deliberation. According to this hypothesis, structural or functional changes in BLA that occur with aging would thus exert their influence on intertemporal choice through a putative 'deliberation circuit', perhaps involving BLA projections to prefrontal cortex (PFC; *Burgos-Robles et al., 2017*). The BLA contains co-distributed neurons that send distinct efferent projections to PFC and NAc (*Pérez-Jaranay and Vives, 1991*; *Ambroggi et al., 2008*; *Shiflett and Balleine, 2010*; *Dilgen et al., 2013*; *McGarry and Carter, 2017*). These distinct populations of BLA efferents may subserve unique roles in intertemporal choice, and further, may be differentially susceptible to aging. In other words, attenuated impulsive choice in aged rats might reflect a failure to engage a BLA-NAc 'outcome evaluation' circuit in combination with a hypoactive BLA-PFC 'deliberation' circuit that mimics the effect of BLA inactivation during deliberation. Future experiments applying circuit-based optogenetic approaches to the study of decision making in aging should be helpful for further elucidating the neurobiological substrates of age-associated alterations in intertemporal choice.

## Conclusions

The current experiments demonstrate several unique roles for BLA activity in intertemporal choice. First, these data demonstrate a novel role for BLA in *promoting* impulsive choices when deciding whether to delay gratification. Moreover, these experiments defined a second role for BLA in *attenuating* impulsive choices, by providing negative feedback about the inadequacy of small vs large reward. Notably, this latter role in negative feedback does not extend to the aversive properties of the delay, indicating that the integrated valuation of reward and costs occurs outside of the BLA. Finally, the current experiments demonstrate that these temporally distinct roles of BLA in decision making change in aging. Specifically, aged rats do not appear to use BLA in any form of outcome evaluation. Moreover, the effects of age on intertemporal choice are mimicked by inactivation of

BLA during deliberation. These findings suggest complex effects of aging within and/or outside BLA, which may uniquely impact distinct BLA efferent circuits. This study is among the first to apply optogenetic techniques to the study of cognitive aging. The findings offer unique insights into how BLA mediates intertemporal choice, and show that optogenetic approaches can be used to complement and extend our understanding of how changes in neural activity guide behavioral alterations in aged subjects.

## Materials and methods

### Subjects

Young (6 months old, n = 24) and aged (24 months old, n = 19 male Fischer 344 x Brown Norway F1 hybrid (FBN) rats were obtained from the National Institute on Aging colony (Charles River Laboratories) and individually housed in the Association for Assessment and Accreditation of Laboratory Animal Care International-accredited vivarium facility in the McKnight Brain Institute building at the University of Florida in accordance with the rules and regulations of the University of Florida Institutional Animal Care and Use Committee and National Institutes of Health guidelines. The facility was maintained at a consistent temperature of 25° with a 12 hr light/dark cycle (lights on at 0600) and free access to food and water except as otherwise noted. Rats were acclimated in this facility and handled for at least one week prior to initiation of any procedures. A subset of rats completed only some of the behavioral epochs due to lost headcaps and premature death, and some rats were excluded entirely for misplaced injections. Only the final numbers of rats included in each analysis are provided below.

### Surgical procedures

Surgical procedures were performed as in our previous work (*Orsini et al., 2017*). Rats were anesthetized with isoflurane gas (1–5% in $O_2$) and secured in a stereotaxic frame (David Kopf). An incision along the midline over the skull was made and the skin was retracted. Bilateral burr holes were drilled above the BLA and five additional burr holes were drilled to fit stainless steel anchoring screws. Bilateral guide cannulae (22-gauge, Plastics One) were implanted to target the BLA (anteroposterior (AP): −3.25 mm from bregma, mediolateral (ML): ±4.95 mm from bregma, dorsoventral (DV): −7.3 mm from the skull surface) and secured to the skull using dental cement. A total of 0.6 µL of a $3.5 \times 10^{12}$ vg/ml titer solution (University of North Carolina Vector Core) containing AAV5 packaged with either halorhodopsin (CamKIIα-eNpHR3.0-mCherry, n = 8 young and n = 7 aged rats) or mCherry alone (CamKIIα-mCherry, n = 4 young and n = 4 aged rats) was delivered through the implanted cannulae at a rate of 0.6 µL per min. Stainless steel obdurators were placed into the cannulae to minimize the risk of infection. Immediately after surgery, rats received subcutaneous injections of buprenorphine (1 mg/kg) and meloxicam (2 mg/kg). Buprenorphine was also administered 24 hr post-operation, and meloxicam 48–72 hr post-operation. A topical ointment was applied as needed to facilitate wound healing. Sutures were removed after 10–14 days and rats recovered for at least 2 weeks before food restriction and behavioral testing began.

### In vitro electrophysiology

For in vitro electrophysiological verification of functional halorhodopsin (eNpHR3.0), young (n = 4) and aged (n = 3) rats underwent surgery as described above except that neither guide cannulae nor skull screws were implanted. Following a 3–4 week survival time, rats were deeply anesthetized via i.p. injection of ketamine (75–100 mg/kg) and xylazine (5–10 mg/kg). The brain was rapidly cooled via transcardial perfusion with cold oxygenated sucrose-laden artificial cerebrospinal fluid (ACSF) containing (in mM): 205 sucrose, 10 dextrose, 1 $MgSO_4$, 2 KCl, 1.25 $NaH_2PO_4$, 1 $CaCl_2$, and 25 $NaHCO_3$. Rats were then decapitated, brains extracted and coronal slices (300 µm) prepared using a Leica VT 1000 s vibratome. Slices were incubated for 30 min at 37°C in oxygenated low-calcium ACSF containing (in mM): 124 NaCl, 10 dextrose, 3 $MgSO_4$, 2.5 KCl, 1.23 $NaH_2PO_4$, 1 $CaCl_2$, and 25 $NaHCO_3$, after which they were transferred to room temperature for a minimum of 30 min prior to experimentation. During recording experiments, slices were bathed in ACSF containing (in mM): 125 NaCl, 11 dextrose, 1.5 $MgSO_4$, 3 KCl, 1.2 $NaH_2PO_4$, 2.4 $CaCl_2$, and 25 $NaHCO_3$, maintained at 28–30°C. The pipette (internal) solution contained (in mM): 125 K-gluconate, 10 phosphocreatine, 1

MgCl$_2$, 10 HEPES, 0.1 EGTA, 2 Na$_2$ATP, 0.25 Na$_3$GTP, and 5 biocytin, adjusted to pH 7.25 and 295 mOsm with KOH. BLA neurons were visualized using a combination of IR-DIC and epifluorescence microscopy using an Olympus BX51WI microscope and a TTL-controlled light source (X-Cite 110 LED light source, XF102-2 filter set, Omega Optical, excitation 540–580 nm, emission 615–695 nm, also used for in vitro activation of eNpHR3.0 for 1 or 4 s). Patch pipettes were prepared with a Flaming/Brown type pipette puller (Sutter Instrument, P-97) from 1.5 mm/0.8 mm borosilicate glass capillaries (Sutter Instruments) and pulled to an open tip resistance of 4–7 MΩ using internal solution and ACSF noted above. Electrophysiological recordings were performed using a Mutliclamp 700B amplifier and Digidata 1440A digitizer (Axon Instruments/Molecular Devices). Electrophysiological data were collected at 20 kHz and low-pass filtered at 2 kHz.

At the conclusion of experiments, a subset of slices was transferred to 10% formalin (4°C, 24 hr) to allow for *post hoc* histological analysis. Slices were washed in PBS, permeabilized in PBS containing 0.1% Triton-X, and incubated in streptavidin conjugate with fluorophore (1:1000, 594 nm, ThermoFisher S32356). Slices were then mounted onto slides and coverslipped using VECTASHIELD. Morphological reconstruction was achieved by creating an all-in-focus maximum intensity projection of a Z-series acquired with a two-photon laser scanning epifluoresence microscope (810 nm excitation).

## Behavioral testing procedures

### Apparatus

Testing was conducted in four identical standard rat behavioral test chambers (Coulbourn Instruments) with metal front and back walls, transparent Plexiglas side walls, and a floor composed of steel rods (0.4 cm in diameter) spaced 1.1 cm apart. Each test chamber was housed in a sound-attenuating cubicle and was equipped with a custom food pellet delivery trough fitted with a photobeam to detect head entries (TAMIC Instruments) located 2 cm above the floor and extending 3 cm into the chamber in the center of the front wall. A nosepoke hole equipped with a 1.12 W lamp for illumination was located directly above the food trough. Food rewards consisted of 45 mg grain-based food pellets (PJAI; Test Diet, Richmond, IN, USA). Two retractable levers were positioned to the left and right of the food trough (11 cm above the floor). A 1.12 W house light was mounted near the top of the rear wall of the sound-attenuating cubicle. For optical activation of eNpHR3.0, laser light (561 nm, 8–10 mW output at the fiber tip, Shanghai Laser and Optics Century) was delivered through a patch cord (200 μm core, Thor Labs) to a rotary joint (1 × 2, 200 μm core, Doric Lenses) mounted above the operant chamber. At the rotary joint, the light was split into two outputs. Tethers (200 μm core, 0.22 NA, Thor Labs) connected these outputs to the bilateral optic fibers (200 μm core, 0.22 NA, 8.3 mm in length; Precision Fiber Products) implanted in the BLA (*Orsini et al., 2017*). A computer running Graphic State 4.0 software (Coulbourn Instruments) was used to control the behavioral apparatus and laser delivery, and to collect data.

### Behavioral shaping and initial training

The intertemporal choice task was based on a design by *Evenden and Ryan (1996)* and was used previously to demonstrate age-related alterations in decision making in both Fischer 344 (*Simon et al., 2010*) and FBN (*Hernandez et al., 2017*) rats (*Figure 1*). Prior to commencement of behavioral testing, both young and aged rats were food-restricted to 85% of their free-feeding weights over the course of 5 days and maintained at these weights for the duration of the experiments. Rats were initially shaped to lever press to initiate delivery of a food pellet into the food trough and were then trained to nosepoke to initiate lever extension. Each nosepoke initiated extension of either the left or right lever (randomized across pairs of trials), a press on which yielded a single food pellet. After two consecutive days of reaching criterion performance (45 presses on each lever), rats began testing on the intertemporal choice task.

### Intertemporal choice task

Each 60 min session consisted of 3 blocks of 20 trials each. The trials were 60 s in duration and began with a 10 s illumination of both the nosepoke port and house light. A nosepoke into the port during this time extinguished the nosepoke light and triggered lever extension. Any trials on which rats failed to nosepoke during this 10 s window were scored as omissions. Each 20-trial block began

with two forced choice trials, in which either the right or left lever was extended, in order to remind rats of the delay contingencies in effect for that block. These forced choice trials were followed by 18 free choice trials, in which both levers were extended. For all trials, one lever (either left or right, counterbalanced across age groups) was always associated with immediate delivery of one food pellet (the small reward), and the other lever was associated with three food pellets (the large reward) delivered after a variable delay. Lever assignment (small or large reward) remained consistent throughout testing. Within a session, the duration of the delay preceding large reward delivery increased across the three blocks of trials. The actual delay durations were adjusted individually for each rat, such that the percent choice of the large reward corresponded to roughly 100% in block 1, 66% in block 2, and 33% in block 3. On all trials, rats were given 10 s to press a lever, after which the levers were retracted, and food was delivered into the food through. If rats failed to press a lever within 10 s, the levers were retracted, lights were extinguished, and the trial was scored as an omission. An inter-trial interval (ITI) followed either food delivery or an omitted trial, after which the next trial began.

Rats were initially trained for 15 sessions on the intertemporal choice task. They were then lightly anesthetized and optic fibers (Precision Fiber Products) were inserted into the guide cannulae such that they extended 1 mm past the end of the guide cannulae and were cemented in place. After recovery, rats resumed training but were now tethered to the rotary joint.

## Evaluation of optogenetic inactivation during specific task epochs

The effects of temporally-discrete optogenetic inhibition of BLA were tested in both young and aged rats using a within-subjects design. Data from sessions occurring just prior to inactivation sessions (in which rats did not receive light delivery) were used as the baseline against which to compare the effects of BLA inactivation. Task epochs in which the BLA was inactivated included: *deliberation* (light delivery began 500 ms prior to illumination of the nosepoke light and continued until the rat pressed the lever, for a maximum of 10 s); *small reward delivery* (light delivery began when food was dispensed and remained on for 4 s); *large reward delivery* (light delivery began when food was dispensed and remained on for 4 s), *delay* (light delivery began upon pressing the large reward lever and remained on throughout delay intervals ranging between 2–24 s); *large reward delivery + delay* (light delivery began upon pressing the large reward lever and remained on until 4 s after the large reward was dispensed), *intertrial interval (ITI*; light delivery began 14 s after reward was dispensed and continued for 4 s). Finally, the sequence of BLA inactivation sessions during each task epoch was counterbalanced across rats.

## Vector expression and cannula placement histology

After completion of behavioral testing, rats were administered a lethal dose of Euthasol (sodium pentobarbital and phenytoin solution; Virbac, Fort Worth, TX, USA) and perfused transcardially with a 4°C solution of 0.1M phosphate buffered saline (PBS), followed by 4% (w/v) paraformaldehyde in 0.1M PBS. Brains were removed and post-fixed for 24 hr then transferred to a 20% (w/v) sucrose solution in 0.1M PBS for 72 hr (all chemicals purchased from Fisher Scientific, Hampton, NH, USA). Brains were sectioned at 35 µm using a cryostat maintained at −20 °C. Sections were rinsed in 0.1M TBS and incubated in blocking solution consisting of 3% normal goat serum, 0.3% Triton-X-100 in 0.1M TBS for 1 hr at room temperature. Sections were then incubated with rabbit anti-mCherry antibody (ab167453, Abcam, Cambridge, MA, USA) diluted in blocking solution at a dilution of 1:1000 (72 hr, 4°C). Following primary incubation, sections were rinsed in 0.1M TBS and incubated in blocking solution containing the secondary antibody (donkey anti-rabbit conjugated to AlexaFluor-488, 1:300) for 2 hr at room temperature. After rinsing in 0.1M TBS, sections were mounted on electrostatic glass slides and coverslipped using Prolong Gold containing DAPI (Thermo Fisher Scientific, Waltham, MA, USA). Slides were sealed with clear nail polish and sections were visualized at 20X using an Axio Imager 2 microscopy system (Carl Zeiss Microscopy, LLC, Thornwood, NY, USA) to assess mCherry expression in BLA neurons. Cannula placements and mCherry expression were mapped onto plates adapted from the rat brain atlas of *Paxinos and Watson (2005)*.

## Experimental design and statistical analysis

### Evaluation of age differences in halorhodopsin effects on BLA neuronal activity

Data analysis was performed using OriginLab and custom electrophysiology analysis software written by CJF. Electrophysiological measures were compared between young and aged cells (obtained from n = 2–4 rats per age) using an independent samples t-test.

### Evaluation of age differences in intertemporal choice under baseline conditions

Raw data files were extracted using a Graphic State 4.0 analysis template that was custom-designed to extract the number of lever presses on each lever (large or small rewards) during forced and free choice trials in each trial block. First, age differences in intertemporal choice performance were tested by analyzing the actual delays used to achieve the target 100%, 66% and 33% choice of the large reward in blocks 1, 2 and 3, respectively. Actual delays were compared using a mixed-factor ANOVA, with age (two levels: young and aged) as the between-subjects factor and block (three levels: blocks 1–3) as the within-subjects factor. Second, the *choice indifference point*, or the delay at which a rat showed equivalent choice of the small and large reward, was calculated and compared between young and aged rats. Choice indifference points were calculated by fitting a trend line to each rat's percent choice of the large reward at each delay block. The slope-intercept formula, y = mx + b (where 'y' is percent choice of the large reward, and 'x' is delay), was then used to solve for the number of seconds (x) at which y = 50% choice of the large reward (the delay at which the rat was equally likely to choose the large or small reward). Choice indifference points were compared between young and aged rats using an independent samples t-test. Alpha was set to 0.05 for all statistical analyses. For statistical results that reached the threshold for significance, $\eta_p^2$ (partial eta squared) and Cohen's *d* were used to report the effect size for mixed-factor ANOVAs and t-tests, respectively, and 1-β was used to report the observed power.

### Evaluation of BLA inactivation on intertemporal choice

Power analyses based on data from an initial cohort of rats (n = 3) were used to determine sample sizes necessary to evaluate the effects of BLA inactivation on choice behavior. These analyses indicated the presence of large effect sizes (greater than 1.0), and that n = 6 rats should be sufficient to detect effects of BLA inactivation, with a power to detect significant differences of 0.95. The effects of light delivery were tested separately for each task epoch (*deliberation, small reward delivery, large reward* delivery, *delay, delay +large reward delivery,* and *ITI*). For each epoch, comparisons were made using a mixed factor ANOVA (laser condition × age × delay block), with age as the between-subjects factor (two levels: young and aged), and laser condition (two levels: laser on or off) and delay block (three levels: delay blocks 1–3) as within-subjects factors. To better understand significant main effects or interactions, *post hoc* analyses were conducted in each age group separately using a repeated-measures ANOVA (laser condition × block). Note that for those epochs in which effects of BLA inactivation during the delay were tested, data analyses were confined to blocks 2 and 3, as there was no delay in block 1.

### Evaluation of choice strategy resulting from BLA inactivation

Additional analyses were conducted to better understand the shifts in choice performance following BLA inactivation during the deliberation and small reward epochs. Graphic State 4.0 templates were created to assess trial-by-trial choices during baseline and BLA inactivation sessions for the deliberation and small reward epochs. Trials were categorized based on choices made on the previous trial. For the deliberation epoch, trials were categorized as 'small-<u>shift</u>-to-large' or 'large-<u>stay</u>-on-large'. For the small reward delivery epoch, trials were categorized as 'large-<u>shift</u>-to-small' or 'small-<u>stay</u>-on-small'. The number of trials in each category was divided by the total number of trials in that session and expressed as a percentage. For each task epoch, percentages of trials in each category were compared using a mixed-factor ANOVA with age as the between-subjects factor (two levels: young and aged) and laser condition as the within-subjects factor (two levels: laser on or off).

## Effects on other task performance measures resulting from BLA inactivation

Other task measures were compared between BLA inactivation and baseline conditions in task epochs in which BLA inactivation produced significant changes in choice behavior. Specifically, on free choice trials, response latency (the time between lever extension and a lever press) was compared. Previous work shows that response latencies can differ for large and small reward levers (*Hernandez et al., 2017*) and hence analyses were conducted separately for each lever using data from delay block 2, during which rats made roughly equivalent numbers of choices on each reward lever. Response latency and total number of trials completed were compared using a mixed factor ANOVA (laser condition × age).

## Acknowledgements

We thank Vicky S Kelly, Shannon C Wall, and Bonnie I McLaurin for assistance with surgeries and behavioral testing, Wanhui Sheng for assistance with two-photon microscopy, and Jeff Thinschmidt for assistance with patch clamp recording. Supported by R01AG029421 (JLB), RF1AG060778 (JLB, BS, CJF), the McKnight Brain Research Foundation (JLB), a McKnight Predoctoral Fellowship and the Pat Tillman Foundation (CMH), and a Thomas H. Maren Fellowship and K99DA041493 (CAO).

## Additional information

### Funding

| Funder | Grant reference number | Author |
|---|---|---|
| McKnight Brain Research Foundation | | Jennifer L Bizon |
| National Institutes of Health | R01AG029421 | Jennifer L Bizon |
| McKnight Foundation | | Caesar M Hernandez |
| Pat Tillman Foundation | | Caesar M Hernandez |
| Thomas H. Maren Foundation | | Caitlin A Orsini |
| National Institutes of Health | K99DA041493 | Caitlin A Orsini |
| National Institutes of Health | RF1AG060778 | Charles J Frazier<br>Barry Setlow<br>Jennifer L Bizon |

The funders had no role in study design, data collection and interpretation, or the decision to submit the work for publication.

### Author contributions

Caesar M Hernandez, Conceptualization, Data curation, Formal analysis, Investigation, Methodology, Writing—original draft, Writing—review and editing; Caitlin A Orsini, Conceptualization, Formal analysis, Investigation, Methodology, Writing—review and editing; Chase C Labiste, Alexa-Rae Wheeler, Tyler W Ten Eyck, Matthew M Bruner, Todd J Sahagian, Scott W Harden, Investigation; Charles J Frazier, Conceptualization, Software, Formal analysis, Supervision, Methodology, Project administration, Writing—review and editing; Barry Setlow, Jennifer L Bizon, Conceptualization, Resources, Supervision, Funding acquisition, Methodology, Writing—original draft, Project administration, Writing—review and editing

### Author ORCIDs

Caesar M Hernandez (iD) https://orcid.org/0000-0001-9690-5119
Scott W Harden (iD) http://orcid.org/0000-0002-0757-1979
Barry Setlow (iD) https://orcid.org/0000-0001-9133-9445
Jennifer L Bizon (iD) https://orcid.org/0000-0002-9517-5844

## Ethics

Animal experimentation: This research was conducted in accordance with the rules and regulations of the University of Florida Institutional Animal Care and Use Committee (protocol number 201604961) and National Institutes of Health guidelines.

## Decision letter and Author response

Decision letter https://doi.org/10.7554/eLife.46174.023
Author response https://doi.org/10.7554/eLife.46174.024

# Additional files

## Supplementary files

• Transparent reporting form
DOI: https://doi.org/10.7554/eLife.46174.021

## Data availability

All data generated or analyzed during this study are included in the manuscript and supporting files.

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
