## [Decision Letter]

Thank you for submitting your article "Aging alters the role of basolateral amygdala in intertemporal choice" for consideration by *eLife*. Your article has been reviewed by Michael Frank as the Senior Editor, Geoffrey Schoenbaum as the Reviewing Editor, and two reviewers. The reviewers have opted to remain anonymous.

The reviewers have discussed the reviews with one another and the Reviewing Editor has drafted this decision to help you prepare a revised submission.

Summary:

In this study, the authors build on prior work regarding the role of the amygdala in an inter temporal choice task to examine effects of aging on its involvement. Young and aged rats were presented with trials involving a choice between two levers, one producing a small reward immediately and another producing a larger reward at a delay, with the length of delay increasing in three steps across trial blocks. They found that aged rats were less sensitive to delay than young controls. After adjusting the delay to obtain similar choice performance in each block between young and old, which was clever, they then ontogenetically inactivated basolateral amygdala in specific time periods of the trials. They found that inactivation during deliberation had similar effects in the two groups, whereas inactivation during evaluation of the small reward affected young but not old rats. Inactivation during a number of other periods did not alter the behavior in either group.

Essential revisions:

Overall the reviewers thought the study was well-designed and analyzed and the results were clear. However, the main concern was that the novelty with regard to aging was predicated on a single null finding, with inactivation in most time periods having apparently no impact. The reviewers felt that negative results in such a cleanly designed study were worthwhile, but that the overall picture was somewhat lost in the discussion. To correct this, there were three essential revisions that seemed important in the Discussion section. One was to consider better the import of the data overall, the limitation of the null result and its significance alongside all the other negative findings. Does this mean the effects of aging are quite subtle? Why is amygdala not involved in these other periods? A second related revision is to conduct the explicit direct age group comparisons of data as requested by R2. A third is to provide some additional detail or discussion of the effectiveness of the optogenetics in young versus aged rats – can expression and the results of the slice physiology be better quantified to show it is quantitatively similar? If revisions along these lines can be made, the reviewers thought this would be an important contribution to the aging literature.

Minor points: see reviews. The title might also be adjusted to reflect the balance of negative results. Also the species used might be mentioned.

*Reviewer #1:*

This report from Hernandez et al., uses a combination of optogenetic methods and parametric task manipulations to explore how basolateral amygdala (BLA) circuitry might contribute to the effects of aging on decision making. The results demonstrate that, whereas BLA inhibition when rats are deliberating about an impending choice enhances delayed gratification independent of age, the same manipulation after reward delivery, during outcome evaluation, shifts decision making in the opposite direction only in young subjects. This pattern suggests that BLA recruitment mediates multiple, temporally discrete processing functions in decision making that are differentially sensitive to aging. The paper is well written, the approach is methodologically rigorous, and the analysis appears appropriate. The findings with respect to BLA contributions to intertemporal choice are novel, but conclusions regarding the effects of aging seem less compelling.

The key findings of the paper regarding aging are illustrated in Figure 5. They demonstrate that BLA inhibition during choice deliberation enhances delayed gratification similarly in young and aged rats (Figure 5A). However, only young rats display an effect of inhibition delivered during choice outcome evaluation (Figure 5B). Evidence that the role of BLA is altered in aging therefore hinges on a single negative result, i.e., the absence of a modulatory effect of BLA inhibition seen in young rats. What would strengthen the conclusions is evidence pointing to the potential direction of effect. Is outcome evaluation insensitive to BLA inhibition because the role of the BLA is altered in aged animals, or is BLA inhibition ineffective because choice outcome evaluation is altered in aging? Viewed in a different way, is 'choice indifference' at a ~5 second delay in young rats really the same processing function as 'choice indifference' at a ~20 second delay in aged rats? Disentangling these alternatives on the basis of the negative result reported seems problematic.

The body of the report indicates that a variety of task-specific performance measures failed to differ as a function of age, but the Tables listing the results do not indicate that direct age group comparisons were conducted (i.e., they reference effects of inactivation and task condition, but not age). Of particular note, response latencies (Table 2) seem generally faster for the aged rats and it would be interesting to know if this effect was reliable.

The results of the in vitro slice physiology provide valuable confirmation, but subjectively they suggest that firing rates after optogenetic inhibition may increase relative to baseline (see pre/post trace in Figure 2D). Are the data sufficient to assess whether this effect is reliable, or age-dependent?

The Materials and methods section mention food restriction, but specifics do not seem to be reported, including potential differential restriction in the young and aged rats.

*Reviewer #2:*

This manuscript by Hernandez and colleagues addresses the hypothesis that age-related increases in the ability to delay gratification are related to changes in the function of the basolateral amygdala (BLA), a region that supports reward-cost evaluation. The investigators used optogenetics to manipulate the BLA in an intertemporal choice paradigm. In this paradigm, rats have a deliberation period followed by the opportunity to choose between a response that will provide a small, immediate reward or a response that will provide a large, delayed reward. If the BLA was inactivated during deliberation, young and aged rats were less impulsive in their choices and were more likely to choose the large delayed rewards. If the BLA was inactivated during outcome evaluation of small rewards, the young rats more impulsive on the next choice and less likely to choose the large, delayed reward. Aged rats were not affected by inactivation during evaluation of small rewards. Choice behavior was not impacted by inactivation during delay, during large outcome evaluation, or delay plus large outcome evaluation. This shift in opposite directions following inactivation of the BLA has been observed in young rats in a similar paradigm in a study conducted by the same group.

The authors interpretation is that the BLA is involved only in certain temporal components of intertemporal decision making under certain reward and that alterations in the recruitment of the BLA intertemporal decision making underlies age-related preferences for delayed gratification.

What is new here? We already know that aging alters delayed gratification and that this relies on the BLA. So, it's not clear what is novel here, unless it is the impact on particular components of the behavior.

One issue is whether differences in the duration of light might underlie some of these effects or underlie some of the null effects. There are some reports that constant NpHR light can induce increases in intracellular chloride accumulation that result in increased excitability of neuronal networks or a rebound effect. So, it is possible that differences in the duration of light could have different effects on the BLA in ways that are different across conditions or different across age groups. The slice work seems to have only addressed the impact of 500 milliseconds of light, so does not address this issue. I have thought about this and I don't think it is a problem. Should it be addressed in the Discussion section?

Although an interesting finding consistent with what we know about aging and the BLA, this paper would have been more impressive if the investigators had moved the aged rats in the other direction, in other words, correcting the age-related enhanced ability to delay gratification.

The trial by trial analysis (Figure 5C and subsection “Altered choice strategy resulting from BLA inactivation during the deliberation and small reward epochs”) was conducted only for the young groups. It would be interesting to see this analysis for the old rats, even though there were only effects for the deliberation epoch.

---

## [Author Response]

Essential revisions:Overall the reviewers thought the study was well-designed and analyzed and the results were clear. However, the main concern was that the novelty with regard to aging was predicated on a single null finding, with inactivation in most time periods having apparently no impact. The reviewers felt that negative results in such a cleanly designed study were worthwhile, but that the overall picture was somewhat lost in the discussion. To correct this, there were three essential revisions that seemed important in the Discussion section. One was to consider better the import of the data overall, the limitation of the null result and its significance alongside all the other negative findings. Does this mean the effects of aging are quite subtle? Why is amygdala not involved in these other periods?

In response to this comment, we have significantly revised and restructured the discussion, with the goal of more clearly highlighting both the novelty of the work and the importance of the null effects.

Why are there so many null results and why is the BLA not involved in the other task periods?

Electrophysiological recordings and neuroimaging of neural activity during complex tasks reliably demonstrate distinct relationships between neural activity and behavior during temporally-specific epochs of task performance (e.g., Setlow et al., 2003; Peters and Buchel, 2011; Sackett et al., 2019). As such, temporally-discrete optogenetic manipulations of neural activity during decision making (in which multiple cognitive operations are being performed) would be expected to produce distinct behavioral effects. Having said that, intertemporal choices are supported by a large network of brain structures and thus it seems unlikely that any single brain structure within this network would contribute in the same way across all of the behavioral epochs as we have defined them within this study (e.g., deliberation vs. delay vs. outcome evaluation). Indeed, one goal of the current study was to define exactly when and how information processed within the BLA guides intertemporal choice behavior. Within this context, the “null” effects offer critical information about when the BLA (and its diverse inputs and outputs) critically mediates intertemporal choice. By extension, the combination of null and positive effects on behavior help define the nature of information processed in this structure that is responsible for driving those choices.

A primary example of the new information that has been gleaned using this optogenetic approach is related to the role of BLA in “outcome evaluation”. We made the somewhat surprising observation that BLA inactivation during outcome evaluation only impacted subsequent choices when paired with the small, immediate reward. We observed no effect on behavior when BLA was inactivated during the delay or the large reward. We speculate that the role of the BLA during outcome evaluation is restricted to providing negative feedback about the small reward outcome received, and that integration of negative feedback about the delay and large reward outcome must originate in other brain loci. As is described more extensively in the revised Discussion section, a role for BLA in conferring negative feedback about the choice outcome being “less than expected” is supported by other data from a risky decision task in which BLA inactivation selectively influenced behavior on trials associated with punished rewards but not on other trials types in which rats received other outcomes (Orsini et al., 2017). There are a number of other brain regions that might process information about delay interval and the large reward and use that information to guide choice behavior. For example, prefrontal cortex and hippocampus are implicated in both intertemporal choice and maintenance of information during delay periods (Churchwell et al., 2009; Mariano et al., 2009; Abela and Chudasama, 2013; Sonntag et al., 2014; Yates et al., 2014). Manipulating neural activity in one or both of these structures during delays might be expected to impact intertemporal choice. In the revised manuscript, we have expanded the discussion of the null effects and their implications for defining the specific role of BLA in intertemporal choice (Discussion section).

Does this mean the effects of aging (on BLA) are quite subtle?

First, it is important to consider that the effects of aging on intertemporal choice task performance are circumscribed. Aged rats perform procedural elements of the task as well as young, and they do discount delayed rewards to some extent. The effects of age are only on the distribution of choices, such that aged rats discount rewards less steeply than young.

On another level, however, the effects of age on the role of BLA in intertemporal choice are not subtle at all, in that the BLA plays a qualitatively different role in outcome evaluation in aged compared to young rats. Importantly, this difference does not necessarily reflect BLA dysfunction in aged rats but does clearly demonstrate that aged rats are not using information from the BLA in the same way as young to make intertemporal choices.

Finally, although there are copious data indicating that neural activity differs between young and aged subjects, there are very few data in the literature that directly address how such differences in activity contribute to age differences in performance. Thus, we would argue that we (as a field) do not yet know what to expect in terms of the magnitude of behavioral effects that might arise from discrete age differences in neural activity within particular brain regions or circuits.

We have substantially revised the Discussion section in order to offer a better interpretation of both the positive and null effects of BLA inactivation. In particular, we have restructured this section to detail the interpretation of first the young data, followed by the aging-associated changes. We hope that this revision will better frame the unique contributions of this study to the literature and how the null effects can be used to sculpt and restrict the interpretation of the BLA’s role in intertemporal decision making.

A second related revision is to conduct the explicit direct age group comparisons of data as requested by R2.

This is an excellent point and we have now added the age comparison to the trial-by-trial analyses (see revised Figure 5 and subsection “Altered choice strategy resulting from BLA inactivation during the deliberation and small reward epochs”). As is evident in Figure 5B and 5D, the trial-by-trial age comparisons are consistent with those observed in the analyses of mean choice of the large reward across trial blocks (Figure 5A and 5C).

A third is to provide some additional detail or discussion of the effectiveness of the optogenetics in young versus aged rats – can expression and the results of the slice physiology be better quantified to show it is quantitatively similar?

We appreciate these comments and have now included an explicit and quantitative comparison of slice electrophysiological data in young and aged rats, including results from new experiments that assessed the effects of light durations that match those used during the behavioral experiments (Results section and Figure 2, as well as Discussion section). Across all experiments, the data show that the effects of optogenetic inactivation are no different in young and aged rats.

If revisions along these lines can be made, the reviewers thought this would be an important contribution to the aging literature.

Reviewer #1:

This report from Hernandez et al., uses a combination of optogenetic methods and parametric task manipulations to explore how basolateral amygdala (BLA) circuitry might contribute to the effects of aging on decision making. The results demonstrate that, whereas BLA inhibition when rats are deliberating about an impending choice enhances delayed gratification independent of age, the same manipulation after reward delivery, during outcome evaluation, shifts decision making in the opposite direction only in young subjects. This pattern suggests that BLA recruitment mediates multiple, temporally discrete processing functions in decision making that are differentially sensitive to aging. The paper is well written, the approach is methodologically rigorous, and the analysis appears appropriate. The findings with respect to BLA contributions to intertemporal choice are novel, but conclusions regarding the effects of aging seem less compelling.The key findings of the paper regarding aging are illustrated in Figure 5. They demonstrate that BLA inhibition during choice deliberation enhances delayed gratification similarly in young and aged rats (Figure 5A). However, only young rats display an effect of inhibition delivered during choice outcome evaluation (Figure 5B). Evidence that the role of BLA is altered in aging therefore hinges on a single negative result, i.e., the absence of a modulatory effect of BLA inhibition seen in young rats. What would strengthen the conclusions is evidence pointing to the potential direction of effect. Is outcome evaluation insensitive to BLA inhibition because the role of the BLA is altered in aged animals, or is BLA inhibition ineffective because choice outcome evaluation is altered in aging? Viewed in a different way, is 'choice indifference' at a ~5 second delay in young rats really the same processing function as 'choice indifference' at a ~20 second delay in aged rats? Disentangling these alternatives on the basis of the negative result reported seems problematic.

To address this issue, we conducted additional analyses in which aged rats were subdivided into those with delays comparable to young rats (“aged delay-matched”) and those with delays longer than those of young (“aged delay-unmatched”). We used a three-factor ANOVA to compare these aged sub-groups (with Sub-group as the between-subjects factor and Laser condition and Block as within-subjects factors). Across all behavioral epochs, this analysis revealed no differences between aged sub-groups. Most importantly, there was no difference in choice behavior between the “aged delay-matched” and “aged delay-unmatched” groups on the behavioral epoch in which BLA inactivation produced different results in young and aged rats (statistics from comparisons on small reward outcome delivery: main effect of sub-group and all interactions involving subgroup: all Fs <0.51, ps>.33). The results of these subgroup analyses demonstrate that differences in the actual delay durations between young and aged rats do not appear to account for age differences in the effects of BLA inactivation during small reward outcome evaluation. Rather, these results are consistent with the interpretation that BLA activity during outcome evaluation simply does not influence intertemporal choice in aged rats as it does in young rats. The full statistical analysis for the subgroup analysis in the small reward outcome epoch is now included in the Results section and summarized in the Discussion section. In addition, sub-group comparisons for each task epoch (figures and full statistical results) are included at the end of this letter. Finally, we have considerably revised the discussion and now devote considerable attention to the issue of whether the “failure” of aged rats to use their BLA during outcome evaluation reflects a deficit in BLA function or rather, a difference in how aging influences the cognitive structure (and the neural circuitry) of decision making. This discussion can be found in the Discussion section.

The body of the report indicates that a variety of task-specific performance measures failed to differ as a function of age, but the Tables listing the results do not indicate that direct age group comparisons were conducted (i.e., they reference effects of inactivation and task condition, but not age). Of particular note, response latencies (Table 2) seem generally faster for the aged rats and it would be interesting to know if this effect was reliable.

We apologize for this oversight. The Results section has now been updated to indicate that there were no effects of age or BLA inactivation on any of these measures, and the Tables themselves have been updated to include the statistical comparisons. As now clarified in the Tables and Results section, despite the numerically shorter response latencies in aged compared to young rats, this difference was not statistically reliable.

*The results of the* in vitro *slice physiology provide valuable confirmation, but subjectively they suggest that firing rates after optogenetic inhibition may increase relative to baseline (see pre/post trace in Figure 2D). Are the data sufficient to assess whether this effect is reliable, or age-dependent?*

The reviewer makes an excellent point. As described above in the response to the editor, we have conducted an extensive quantitative comparison of electrophysiological responses to halorhodopsin in young and aged rats. These analyses reveal no differences between young and aged rats (Results section and Figure 2, as well as Discussion section.

The Materials and methods section mention food restriction, but specifics do not seem to be reported, including potential differential restriction in the young and aged rats.

We thank the reviewer for pointing out this oversight. We now include a more detailed description of food restriction procedures in the Materials and methods section, including the fact that the procedures were the same in young and aged rats.

Reviewer #2:

This manuscript by Hernandez and colleagues addresses the hypothesis that age-related increases in the ability to delay gratification are related to changes in the function of the basolateral amygdala (BLA), a region that supports reward-cost evaluation. The investigators used optogenetics to manipulate the BLA in an intertemporal choice paradigm. In this paradigm, rats have a deliberation period followed by the opportunity to choose between a response that will provide a small, immediate reward or a response that will provide a large, delayed reward. If the BLA was inactivated during deliberation, young and aged rats were less impulsive in their choices and were more likely to choose the large delayed rewards. If the BLA was inactivated during outcome evaluation of small rewards, the young rats more impulsive on the next choice and less likely to choose the large, delayed reward. Aged rats were not affected by inactivation during evaluation of small rewards. Choice behavior was not impacted by inactivation during delay, during large outcome evaluation, or delay plus large outcome evaluation. This shift in opposite directions following inactivation of the BLA has been observed in young rats in a similar paradigm in a study conducted by the same group.The authors interpretation is that the BLA is involved only in certain temporal components of intertemporal decision making under certain reward and that alterations in the recruitment of the BLA intertemporal decision making underlies age-related preferences for delayed gratification.What is new here? We already know that aging alters delayed gratification and that this relies on the BLA. So, it's not clear what is novel here, unless it is the impact on particular components of the behavior.

As the reviewer suggests, the novelty of these experiments lies in part with the demonstration that BLA activity drives choice behavior only during temporally discrete components of intertemporal decision making. Specifically, these data reveal an influence of BLA on choice behavior during deliberation between choice options that is opposite of that produced by BLA lesions. Moreover, we show that effects of BLA lesions can be reproduced by inactivation of BLA selectively during small reward outcome evaluation in young rats. These data define multiple, temporally-distinct roles for BLA in intertemporal decision making that influence choice behavior in opposite directions.

A second novel element of these results is that they show that these discrete roles of the BLA in decision making change in aging. We have substantially revised the entire Discussion section in part to better highlight each novel aspects of the results. We further highlight other points of novelty, including that this is one of the first studies to employ optogenetic approaches to study either cognitive aging or intertemporal choice. In reference to these points, it is important to recognize that the data do not simply replicate or verify results from other studies using a host of different techniques (lesion, behavioral pharmacological, electrophysiological, neuroimaging). The optogenetic findings instead extend and offer unique insights into the BLA’s role in intertemporal choice, and show that this approach can be used to complement and extend other approaches used to evaluate neural activity at the cellular and systems level.

One issue is whether differences in the duration of light might underlie some of these effects or underlie some of the null effects. There are some reports that constant NpHR light can induce increases in intracellular chloride accumulation that result in increased excitability of neuronal networks or a rebound effect. So, it is possible that differences in the duration of light could have different effects on the BLA in ways that are different across conditions or different across age groups. The slice work seems to have only addressed the impact of 500 milliseconds of light, so does not address this issue. I have thought about this and I don't think it is a problem. Should it be addressed in the Discussion section?

The reviewer raises several important points. Regarding the effects of age on BLA neurons’ response to optogenetic inactivation, we have now conducted explicit and quantitative comparisons of halorhodopsin effects in young and aged BLA neurons using in vitro electrophysiology. These experiments show equivalent responses to light durations that match those used during behavioral testing (Results section and Figure 2, as well as Discussion section). The reviewer also raised the possibility that differences in light duration could contribute to age differences in the effects of BLA inactivation on behavior. Importantly, age differences were only observed for inactivation during the small reward epoch, in which the duration of light delivery (4 seconds) was equivalent in young and aged rats. Moreover, for the only task epoch (the delay prior to large reward delivery) for which the duration of light delivery did differ between young and aged rats, BLA inactivation had no effect in either age group. We have now made these points more explicit in the Discussion section.

Although an interesting finding consistent with what we know about aging and the BLA, this paper would have been more impressive if the investigators had moved the aged rats in the other direction, in other words, correcting the age-related enhanced ability to delay gratification.

We agree with the reviewer that this is an interesting set of experiments, which are currently in the planning stages in our labs. As these experiments will require several years to complete in both young and aged rats, however, we have elected to make them a focus of future publications.

The trial by trial analysis (Figure 5C and subsection “Altered choice strategy resulting from BLA inactivation during the deliberation and small reward epochs”) was conducted only for the young groups. It would be interesting to see this analysis for the old rats, even though there were only effects for the deliberation epoch.

This is an excellent point and we have now added the age comparison to the trial-by-trial analyses (see revised Figure 5 and subsection “Effects on choice behavior of BLA inactivation during the small reward”). As is evident in Figure 5B and 5D, the trial-by-trial age comparisons are consistent with those observed in the analyses of mean choice of the large reward across trial blocks (Figure 5A and 5C).